

# Characterization of the new BATCH Teflon chamber and on-line analysis of isomeric multifunctional photooxidation products

Finja Löher[1,2], Esther Borrás[3], Amalia Muñoz[3], Anke Christine Nölscher[1,2]

5  [1]Department of Atmospheric Chemistry, University of Bayreuth, 95447 Bayreuth, Germany
[2]Bayreuth Center of Ecology and Environmental Research (BayCEER), University of Bayreuth, 95447 Bayreuth, Germany
[3]Fundación Centro de Estudios Ambientales del Mediterráneo (CEAM), 46980 Paterna, Valencia, Spain

*Correspondence to*: Finja Löher (finja.loeher@uni-bayreuth.de) and Anke C. Nölscher (anke.noelscher@uni-bayreuth.de)

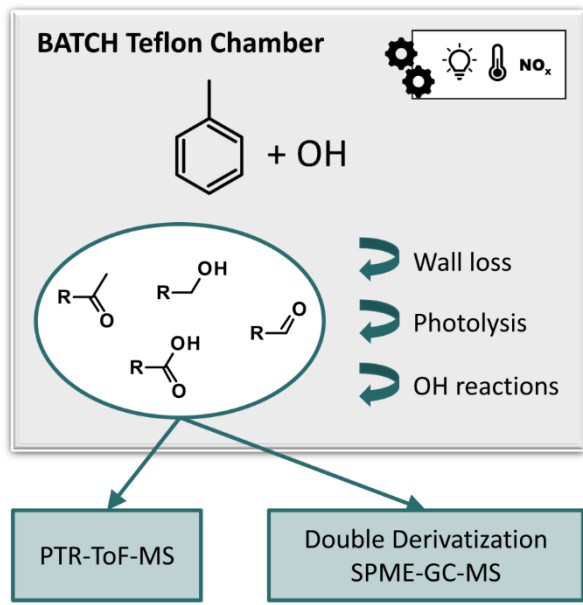



**Abstract.**

The photooxidation of volatile organic compounds (VOCs) in the troposphere has important implications for air quality, weather, and climate. A deeper understanding of the underlying mechanisms can be achieved by studying these reactions under controlled conditions and analyzing the emerging photooxidation products. This requires dedicated laboratory infrastructure as well as sensitive and selective analytical techniques. Here, we constructed a new 300 L indoor Teflon atmospheric simulation chamber as part of the Bayreuth ATmospheric simulation CHambers (BATCH) infrastructure. The chamber was irradiated by a bandpass-filtered solar simulator that enabled experiments with realistic photon fluxes and OH radical concentrations. It was coupled to a proton-transfer-reaction – time-of-flight – mass spectrometer (PTR-ToF-MS) and a solid phase microextraction – gas chromatography – mass spectrometry (SPME-GC-MS) system for the on-line analysis of the precursor VOC and its oxidation products in the gas phase. As part of the SPME-GC-MS method, multifunctional oxygenated compounds (carbonyls, alcohols, carboxylic acids) were derivatized with O-(2,3,4,5,6-Pentafluorobenzyl)-hydroxylamine (PFBHA) and N-trimethylsilyl-N-methyl trifluoroacetamide (MSTFA). We designed a permeation source for the on-line addition of internal standards to improve method reproducibility. The joint setup was tested and validated by studying the OH radical-induced photooxidation of toluene, one of the most abundant aromatic hydrocarbons in the atmosphere. For chamber characterization, we first derived the photolysis rates for several typical toluene products in the irradiated BATCH Teflon chamber ($1.77 \times 10^{-8}$ – $3.02 \times 10^{-4}$ s$^{-1}$). Additionally, wall loss rates were determined empirically ($4.54 \times 10^{-6}$ – $8.53 \times 10^{-5}$ s$^{-1}$), and then parameterized according to fundamental molecular properties. For the cresols and benzyl alcohol, we compiled a weighted calibration factor for the PTR-ToF-MS, taking into account isomer-specific sensitivities as well as the relative distribution as determined by the SPME-GC-MS. The weighted calibration improved the instrumental agreement to 15 %, whereas the PTR-ToF-MS overestimated the sum of the isomers by 25 % compared to the SPME-GC-MS concentrations when using the averaged calibration factor. Thus, the combined data set offered insight into both temporal trends and the isomeric composition. Finally, we conducted six toluene photooxidation experiments to evaluate the ring-retaining first generation products. Based on the loss-corrected concentrations, we derived formation yields for *o*-cresol (8.0±1.8 %), *m*-cresol (0.4±0.1 %), *p*-cresol (2.4±0.6 %), benzyl alcohol (0.5±0.1 %), and benzaldehyde (4.6±1.7 %) under NO$_x$-free conditions at T = 298±1 K. These yields are consistent with previous studies and therefore serve as proof-of-concept for our applied methods.



## 1 Introduction

Volatile organic compounds (VOCs) readily undergo photooxidation in the troposphere. These reactions are tightly linked to the formation of secondary pollutants such as ground-level ozone ($O_3$) or secondary organic aerosol (SOA) (Henze et al., 2008; Kanakidou et al., 2005; Zhao et al., 2022). Furthermore, they directly impact the ambient budget of hydroxyl (OH) radicals and thus the oxidative capacity of the atmosphere (Lelieveld et al., 2008; Williams et al., 2016). Indirectly, VOC photooxidation affects radiative forcing as it competes with the removal of greenhouse gases such as methane ($CH_4$) (von

Schneidemesser et al., 2015). To improve and complement model predictions of air quality and climate, the underlying reaction mechanisms of VOC photooxidation therefore need to be better constrained.

Atmospheric simulation chambers (ASCs) are an established and powerful tool to mimic tropospheric oxidation processes and to derive both mechanistic understanding and kinetic data (Chu et al., 2022; Finlayson-Pitts and Jr, 1999; Kiendler-Scharr et

al., 2023). Typical applications include photooxidant studies (e.g. Carter et al., 1979), gas-phase reaction and product studies (e.g. Zaytsev et al., 2019), and aerosol studies (e.g. Charan et al., 2020). In ASCs, added reagents and ambient conditions can be controlled with great precision. Thus, they enable gradual increases in complexity, the imitation of diverse environments and chemical regimes, and finally transferability to the real world. These data in turn provide the foundation of atmospheric chemistry models such as the near-explicit Master Chemical Mechanism (MCM) for gas-phase reactions (Jenkin et al., 2003;

Saunders et al., 2003).

ASCs worldwide differ in their size and shape, material, and light source. For instance, the EUPHORE (EUropean PHOtoREactor) chamber is spherical, irradiated by natural sunlight, and one of the largest photoreactors in Europe with a volume of about 200 m³ (Muñoz et al., 2018; Zádor et al., 2006). In contrast, other chambers fit into laboratory facilities (e.g.

Carter et al., 2005; Huang et al., 2017). Such indoor chambers have the advantage of being independent of weather and season but require carefully tuned artificial light sources. Knowledge of the spectral distribution and intensity of these lights sources is needed for assessing the analogy to the natural solar spectrum, radical production rates, and photolytic losses of the involved species. Regarding the wall material of ASCs and flow reactors, common choices include borosilicate glass (e.g. Behnke et al., 1988), quartz (e.g. Huang et al., 2017), stainless steel (e.g. Shaw et al., 2018), and Teflon films (e.g. Leskinen et al., 2015).

Teflon films provide flexible arrangements of the chamber design and volume, are UV-transparent, and are a preferred option for many gas-phase studies due to their high chemical inertia (Schwantes et al., 2017; Zádor et al., 2006; Zaytsev et al., 2019). Nevertheless, wall losses matter in Teflon chambers as well, especially for species with low volatility, for small chambers, and for experiments conducted over long time frames (Grosjean, 1985; Krechmer et al., 2020; Ye et al., 2016; Zhang et al., 2015). Regardless of their specific design, all chambers therefore need to be characterized thoroughly (Alfarra et al., 2023; Carter et

al., 2005; Leskinen et al., 2015; Ma et al., 2022) to account not only for contamination and artefacts but also for chamber- and compound-specific losses.



In recent years, the identification and quantification of the multifunctional products formed during VOC photooxidation has become increasingly important and has contributed to great advances in the understanding of autoxidation (e.g. Rissanen, 2021), gas-particle-partitioning (e.g. Gkatzelis et al., 2018), and peroxy radical chemistry (e.g. Berndt et al., 2018). Their analysis is often approached by spectroscopic methods (e.g. Klotz et al., 1998; Olariu et al., 2002) or stand-alone mass spectrometry (MS) (e.g. Baltaretu et al., 2009; Schwantes et al., 2017; Zaytsev et al., 2019). A powerful soft ionization technique for the real-time MS analysis of gas-phase species is by proton transfer reaction (PTR) (Lindinger et al., 1998). When using high resolution mass spectrometers such as Time of Flight (ToF) models, PTR-ToF-MS spectra give insights into sum formulas and enable suspect screening (Romano and Hanna, 2018).

Meanwhile, isomeric information remains scarce for many reaction systems, even though isomers can differ in their properties, rate constants, and yields, and can help elucidating precise reaction mechanisms and intermediate species (Atkinson et al., 1980; Olariu et al., 2002). To distinguish between mass-equal isomers, MS detectors can be coupled to gas chromatography (GC) so that compounds are identified along two dimensions. However, oxygenated products such as ketones, aldehydes, alcohols, and carboxylic acids are often not suitable for GC analysis due to their high fragility and polarity. If analysable at all, there is a substantial risk of losses, poor separation, surface reactions, or conversions to other species during the chromatographic process, resulting in a lowered sensitivity and potentially creating artefacts (Rivera-Rios et al., 2014; Vasquez et al., 2018). To mitigate these problems, Borrás et al. (2021) have recently proposed an on-line GC-MS method with solid phase microextraction (SPME) sampling and on-fibre double derivatization of fragile analytes.

SPME is a sampling and enrichment technique which was first developed by Arthur and Pawliszyn (1990). The associated fibres are coated with a thin layer of a stationary sorbent phase which is exposed to extract and retain analytes during sampling. SPME can be used as a preparatory step to GC-MS analysis, in which case the high temperature in the GC inlet desorbs the analytes from the coated fibre. Advantages include high selectivity and sensitivity, minimization of human error due to the high degree of automation associated with autosampler systems, the reduction in solvent consumption, and the increased analytical throughput by fewer preparation steps and shorter extraction times (Arthur and Pawliszyn, 1990; Borrás et al., 2021; Koziel and Pawliszyn, 2001). However, the performance and reproducibility of SPME can be compromised by saturation effects and competition between analytes due to the limited fibre sorption capacity (Bartelt, 1997) as well as varying extraction efficiencies due to fibre effects (Tumbiolo et al., 2004).

By controlled chemical modification, derivatization procedures can help preserving molecular structures and stabilizing analytes throughout the chromatographic process. Additionally, the compounds of interest can be made more amenable to the specific analytical technique, such as by increasing their volatility for subsequent GC separation. For the analysis of photooxidation products, this concerns in particular the derivatization of hydroxy groups (e.g. alcohols, carboxylic acids) and



carbonyl groups (e.g. aldehydes, ketones). The most common approaches for subsequent GC-MS analysis include silylation for the hydroxy group using reagents like *N*-trimethylsilyl-*N*-methyl trifluoroacetamide (MSTFA) or *N,O*-bis(trimethylsilyl)trifluoroacetamide (BSTFA), and oxime formation for the carbonyl group using for instance *O*-(2,3,4,5,6-Pentafluorobenzyl)-hydroxylamine (PFBHA). These derivatization techniques can also be used in combination to cover a

broad range of compounds with diverse functionalities (Pindado Jiménez et al., 2013; White et al., 2014; Yu et al., 1998). As a means of automating the analytical process, derivatization can be performed directly on SPME fibres, which has been reported for PFBHA previously (Gómez Alvarez et al., 2007; Martos and Pawliszyn, 1998; Schmarr et al., 2008).

Here, we constructed and characterized a new indoor Teflon ASC as part of the **B**ayreuth **AT**mospheric simulation **CH**ambers

(BATCH). For the purpose of studying multifunctional VOC photooxidation products, we coupled the BATCH Teflon chamber to a PTR-ToF-MS and an on-line SPME-GC-MS system with double derivatization using PFBHA and MSTFA. The developed methods were tested and validated by studying the reaction of toluene with OH radicals.

## 2 Study System: Reaction of Toluene with OH Radicals

Toluene is one of the most abundant aromatic hydrocarbons in the atmosphere (Cabrera-Perez et al., 2016). Most of its first

generation products are well-studied (Atkinson et al., 1980; Bloss et al., 2005; Klotz et al., 1998; Zaytsev et al., 2019), making it a suitable and relevant reference system (e.g. Leskinen et al., 2015; Ma et al., 2022). Toluene reacts with OH radicals at a rate of about $k = 5.6 \times 10^{-12}$ molecules$^{-1}$ cm$^3$ s$^{-1}$ (IUPAC, 2024) either via addition of the OH radical to the aromatic ring structure or via H abstraction from the substituted methyl group. The addition pathway is dominant with a branching ratio in the range of $0.85 - 0.93$ (Atkinson et al., 1980; Hu et al., 2007; Wu et al., 2014). The MCMv3.1 (Bloss et al., 2005) distinguishes between

four channels in the primary chemistry of toluene (Fig. 1). Following the formation of an intermediate OH-toluene-adduct (Klotz et al., 1998; Zhang et al., 2019), the addition pathway can lead to ring-retaining cresol isomers (*cresol channel*, 18 % yield) or to ring-opened products such as an epoxydicarbonylene compound (*epoxy-oxy channel*, 10 % yield). Alternatively, peroxy radical isomerization can produce bicyclic peroxy radicals, which ultimately fragment to form the α-dicarbonyl compounds glyoxal and methylglyoxal along with their respective furanone or γ-dicarbonyl co-products (*dicarbonyl channel*,

65 % yield). In the H abstraction pathway (*benzaldehyde channel*, 7 % yield), benzyl radicals readily form peroxy radicals, which undergo a range of bimolecular reactions, producing ring-retaining products such as benzaldehyde and benzyl alcohol (Bloss et al., 2005).

Here, we aimed at evaluating the formation yields of the first generation products that can be calibrated and quantified and

that are specific to their reaction channel. The dicarbonyl products glyoxal and methylglyoxal occur not only in the primary chemistry of toluene but also as secondary products in most channels (Atkinson et al., 1980; Bloss et al., 2005; Wagner et al., 2003). Although their primary production is typically dominant (Gómez Alvarez et al., 2007; Volkamer et al., 2001), a





contribution from secondary sources to their observed concentrations cannot be fully dismissed. Meanwhile, their co-products remain poorly constrained (Bloss et al., 2005; Jenkin et al., 2003; and references therein) and are not available as authentic

standards. The epoxy-oxy channel is highly controversial (e.g. Hu et al., 2007; Wu et al., 2014; Zaytsev et al., 2019) and included in the MCM only based on qualitative indications in order to balance the reaction flux (Bloss et al., 2005; Jenkin et al., 2003). No authentic standard of the epoxydicarbonylene product is available. Therefore, we focus especially on the main $NO_x$-free ring-retaining first generation products of toluene, namely the cresol isomers, benzyl alcohol, and benzaldehyde. For the purpose of evaluating the novel setup, however, we refer to the entire test system with its wide range of multifunctional

products in the first and second generation (beyond the scope of the photooxidation chemistry shown in Fig. 1).

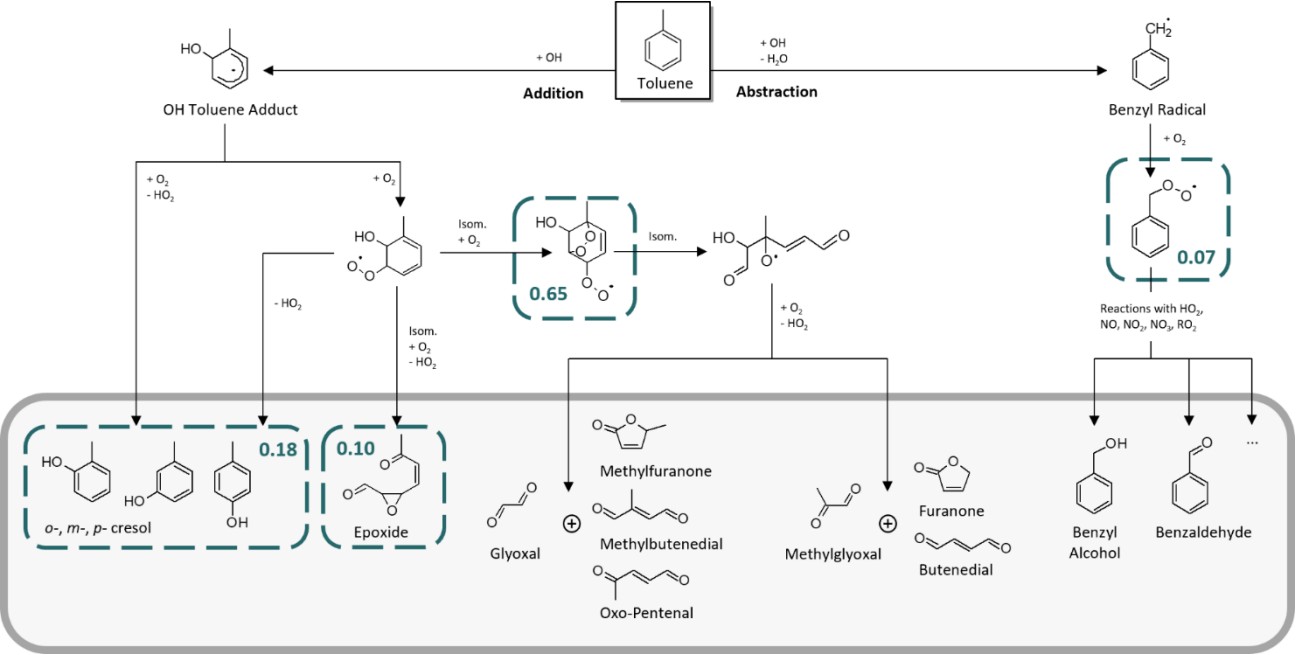

**Figure 1: Primary chemistry of toluene following the reaction with OH radicals in the MCM.** The blue boxes represent the four major channels and their corresponding yields. The grey area includes the main first generation products.




# 3 Experimental

An overview of the novel laboratory setup is provided in Fig. 2. The BATCH Teflon chamber was located in a temperature-controlled room (Hans Zettner GmbH), in which the temperature could be adjusted between -25 °C and 35 °C with a stability

of ±1 K (DeLonghi, HCS 2550 FTS and Dixell, XR170C).

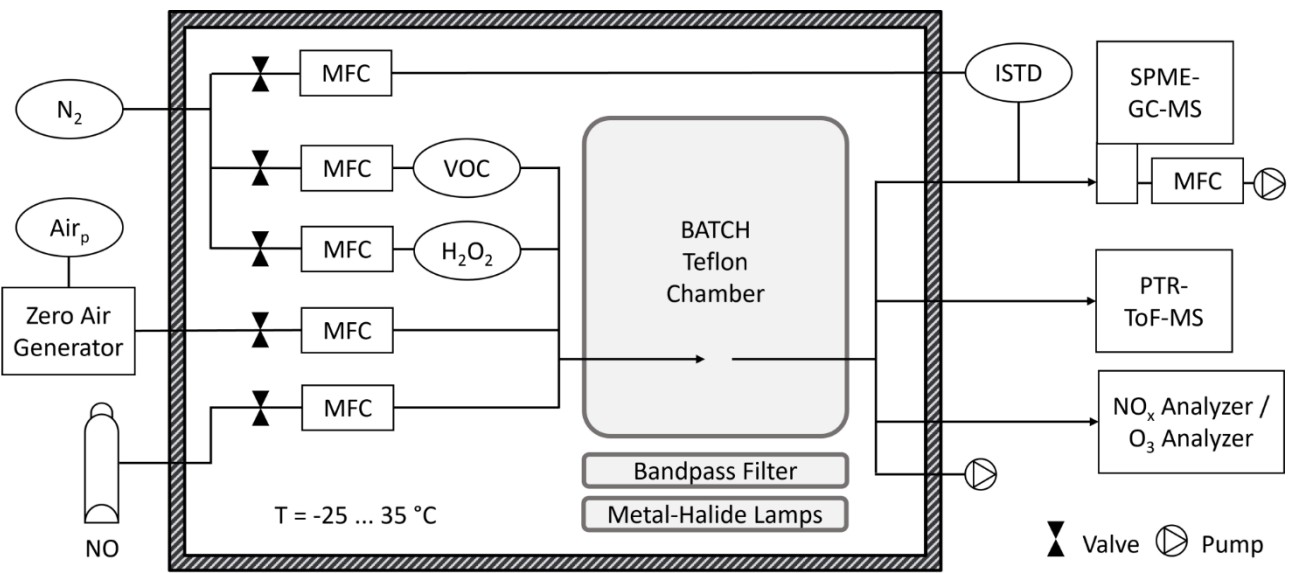

**Figure 2: BATCH Teflon chamber with infrastructure and instrumentation.** The Teflon chamber was installed in a temperature-controlled room (T = -25 °C – 35 °C) and placed above a bandpass-filtered solar simulator. Pressurized air and $N_2$ were available from the in-house gas supply. Added Reagents included the VOC precursor, hydrogen peroxide ($H_2O_2$) as OH radical precursor, optionally nitric
oxide (NO), and zero air as matrix. The VOC precursor and $H_2O_2$ were introduced with a flow of $N_2$. The chamber had one connector which was used as inlet during filling and as outlet during the experiments. All lines were made from polytetrafluoroethylene (PTFE). Coupled analytical instruments included a $NO_x$ analyser, an $O_3$ analyser, a PTR-ToF-MS, and an on-line SPME-GC-MS with attached sampling cell. Two internal standards (ISTDs) were added into the SPME-GC-MS flow using a permeation source. Dimensions are not to scale.

## 3.1 Atmospheric Simulation Chamber

The BATCH Teflon chamber was made from UV-transparent fluorinated ethylene propylene (FEP). We constructed the pillow-shaped bag by folding a single sheet of a 50 μm thick FEP film (Chemours, 200A FEP100) and heat-sealing (Dieck, UM 802) the three open sides. Excluding the heat-sealed areas, the empty chamber was 140×110 cm in size. It could be filled to approximately 300 L, in which case the surface-area-to-volume ratio was about SA:V = 0.1 cm⁻¹. To reduce the risk of

leaks, only one connector was installed at the front side of the chamber and used both as inlet (during filling) and outlet (during ongoing experiments). The chamber was suspended in a metal scaffold, with the two short sides being used as top and bottom ends. It was not further stabilized into a specific shape, so that it remained flexible and fully collapsible, and could be operated in batch mode.



### 3.1.1 Light Source and Photolysis Rates

The chamber was irradiated by a solar simulator, described in detail previously (Bleicher, 2012; Buxmann et al., 2012; Siekmann, 2018; Wittmer et al., 2015). The solar simulator was located underneath the chamber and consisted of seven metal-halide gas discharge medium-arc lamps (Osram, HMI 1200 W, HMI = hydrargyrum medium-arc iodide) with a total power of 8.4 kW. A two-fold bandpass filter was positioned in between the lamps and the chamber to better reproduce the natural solar spectrum. As UV filter, a 3 mm thick borosilicate glass plate (Schott, Tempax) was mounted approximately 1 m above the

lamps. A 3 cm column of decalcified water on top of the glass plate was used both as an IR filter and as a cooling system. During the experiments, the water was circulated to prevent overheating.

The emission spectrum of the solar simulator was already available from a previous measurement by Bleicher (2012). However, aging and replacement of the lamps, changes in the optical properties of the glass plate, or trace contaminations in

the water layer can affect the photon flux. Therefore, we recorded an updated emission spectrum with a spectroradiometer (StellarNet, SILVER-Nova) in the wavelength range $\lambda = 325 – 1000$ nm. The instrument was equipped with a 14 µm slit for a spectral resolution of 0.75 nm, and a CR2 cosine receptor calibrated with NIST traceable calibration lamps. An aperture with 11 % transmission was used to extend the dynamic range. We performed 30 measurements in 15 different positions above the solar simulator to represent the spatial expansion of the inflated chamber. We selected 3 vertical planes (bottom, middle, top)

and 5 spots (middle, front, back, left, right) on each plane. The final spectrum was derived as the average of these measurements. In order to characterize lower wavelengths ranges, we compared our recorded spectrum to the old measurement by Bleicher (2012). We scaled the old spectrum to the new spectrum according to the data available in the spectral range of $\lambda = 350 – 500$ nm. This scaled spectrum was used for the wavelength range $\lambda = 262 – 325$ nm.

We calculated the photolysis rates of the individual compounds according to Eq. (1):

$$J = \int_{\lambda_{min}}^{\lambda_{max}} \phi(\lambda)\sigma(\lambda)F(\lambda)\Delta\lambda \tag{1}$$

where $J$ is the photolysis rate in s⁻¹, $\lambda$ is the wavelength in nm, $\phi$ is the quantum yield in molecules photons⁻¹, $\sigma$ is the absorption cross section in cm² molecules⁻¹, and $F$ is the photon flux of the solar simulator in photons cm⁻² nm⁻¹ s⁻¹. Compound-specific absorption cross sections and quantum yields were taken from the literature (Table S1). Whenever available, IUPAC-

recommended values were selected. If such recommendations were missing, absorption cross section data sets were selected to be recorded at a temperature close to T = 298 K, have a high spectral resolution, and cover a wide wavelength range. If no absorption cross section was available, we assumed that no substantial photolytic loss occurred and did not compile a photolysis rate. For the quantum yields, we used wavelength-specific values or distinct values assigned to a defined actinic range. Otherwise, we assumed a quantum yield of 1 which can be regarded as the upper limit. More details and visualization are

provided in the Supplement S1.1 and Fig. S1.



To validate the recorded spectrum, we performed two actinometric experiments. Firstly, we evaluated the $NO_2$ photolysis rate as described for example by Bohn et al. (2005). We injected 40 ppbv of $NO_2$ (Rießner-Gase, 106 ppmv in $N_2$) into the dark chamber in a zero air matrix. After 15 minutes for mixing, the solar simulator was ignited to start the photochemistry. The

photolysis rate of $NO_2$ was calculated with $NO_2$, NO, and $O_3$ in photostationary state (for details, see Supplement S1.2 and Fig. S2). Secondly, we measured the photolysis rate of methylglyoxal by injecting 120 ppbv of the pure compound into the chamber and observing the wall-loss-corrected decay while the chamber was irradiated (see Supplement S1.2 and Fig. S3). Both $NO_2$ and methylglyoxal absorb in a broad UV-vis range (Table S1) and are hence useful to reference spectral data.

### 3.1.2 Wall Losses

To obtain the individual wall losses, a solution containing all authentic standards was injected into the chamber in a $N_2$ matrix (c = $12.3 \times 10^{10}$ molecules $cm^{-3}$, see SPME-GC-MS calibration in Sect. 3.2.1). We then recorded five consecutive SPME-GC-MS measurements in the dark. For each compound, the relative response was plotted against time and the first order loss rate in $s^{-1}$ was derived by exponential regression. In order to characterize compounds without available authentic standards as well, we developed a parameterization of these empirical rates as a function of their vapor pressure, molecular weight, and oxygen-

to-carbon ratio, as described in more detail in Sect. 4.1.2.

## 3.2 Analytical Instrumentation

While the SPME-GC-MS enabled the quantitative and isomer-resolved analysis of photooxidation products, the PTR-ToF-MS was used for analysing the sum of isomers on a high temporal resolution, for suspect screening, and for monitoring toluene as the precursor VOC. All solvents and authentic standards were purchased in the highest available purity (see Supplement S2.1).

The instrumental setup was complemented by $NO_x$ and $O_3$ analysers (see Supplement S2.2).

### 3.2.1 SPME-GC-MS for Isomer-Resolved VOC Product Analysis

The SPME-GC-MS setup included an SPME autosampler unit (PAL, RSI 85), a gas chromatograph (Agilent, 7890B), and a mass spectrometer (Agilent, 5977A, electron ionization source Xtr 350, Single Quad). The GC-MS was operated with splitless injections, a standard HP-5MS column ramping from a temperature of 45 °C to 280 °C, and in both scan and selected ion

monitoring (SIM) mode. Details on the instrumental settings are listed in Table S2. The selected SPME fibre had a mixed polydimethylsiloxane (PDMS) and divinylbenzene (DVB) coating (Agilent, 5191-5873, PDMS/DVB, 65 µm thickness). Joint PDMS/DVB coatings are bipolar and combine absorption and adsorption extraction mechanisms. They are not only well-suited for the analysis of oxygenated products but also have a high affinity towards nitro-aromatic compounds and towards amines such as PFBHA. We exchanged the SPME fibres every three to four experiments.


We adopted the on-fibre double-derivatization method of Borrás et al. (2021), including (1) the extraction of PFBHA as carbonyl reagent, (2) the on-line sampling of the gas-phase photooxidation products, (3) the extraction of MSTFA as



hydroxy/carboxylic reagent, and (4) the thermal desorption of the derivatized compounds from the fibre into the GC-MS inlet. The derivatization reagents were prepared freshly for each experiment in 20 mL glass vials. 2 mL of PFBHA solution ($c$ = 85

mg $L^{-1}$ in $H_2O$) were made from a concentrated stock solution ($c$ = 850 mg $L^{-1}$ in $H_2O$) which was prepared weekly and stored in the dark at 4 °C. For the silylation reagents, 40 µL of MSTFA were mixed with 10 µL of trimethylchlorosilane (TMCS) as catalyst for higher silylation efficiency. At the beginning of each SPME-GC-MS sequence, the fibre was conditioned at elevated temperature (T = 240 °C) in an inert $N_2$ environment. If a new fibre was used, it was placed in the conditioning cell for 60 minutes. If an old fibre was re-used, it was first cleaned with methanol (3 minutes immersion, 10 minutes desorption)

and then conditioned for 30 minutes. The steps of each subsequent sampling cycle are summarized in Table S3 and were automated via the SPME autosampler unit. Including fibre conditioning times, an entire run took 41 minutes.

A customized PTFE sampling cell (Fig. S4) was placed in the SPME-GC-MS sample tray, enabling the on-line extraction of the photooxidation products (for details, see Supplement S3.2). In the cell, the fibre was placed perpendicular to the sample

air flow for better sampling efficiency (Gómez Alvarez et al., 2012). We used a flow rate to 5 SLPM to accommodate the limited chamber volume and to prevent excessive physical stress on the fibre, while still providing enough mass for analysis and enabling fast equilibration and transportation through the transfer line. The resulting air velocity was 131 cm $s^{-1}$, which is well above the critical value of 10 cm $s^{-1}$ (Gómez Alvarez et al., 2012). The calculated Reynolds number of Re = 760 indicates laminar flow conditions. To maintain comparable conditions between the experiments and to reduce losses, the transfer line

from the chamber to the sampling cell (~ 6 m length, 6 mm inner diameter) was temperature-controlled to 50 °C.

We used two internal standards (ISTDs) to correct for the general variability of SPME fibres (Tumbiolo et al., 2004) as well as for variations in the derivatization efficiency. These ISTDs were added into the transfer line between the chamber and the SPME-GC-MS sampling cell by means of a customized permeation source (for details, see Supplement S3.3 and Fig. S5).

Phenol-$d_6$ was used to correct alcohols and carboxylic acids (derivatized by MSTFA/TMCS), while acetophenone-$d_8$ was used to correct aldehydes and ketones (derivatized by PFBHA). Multifunctional compounds were assigned to the ISTD which resulted in the higher $R^2$ value of the calibration curve. Compounds lacking functional groups amenable to the derivatization scheme were not ISTD-corrected (Table 1).

We tested several known toluene oxidation products. The retention times and SIM masses of the compounds which were included in the final method are provided in Table 1. The SIM masses were selected based on the electron ionization (EI) mass spectra measured for the authentic standards. We considered only those SIM masses which result from mass shifts and fragmentations that are typical for the respective derivatization process (for details, see Supplement S3.4 with Fig. S6 and S7). Thus, we ensured an unambiguous assignment of the quantified ion to the derivatized compound structure. This was necessary

so that the internal standards, which were also specifically monitored in their derivatized form, could provide a meaningful correction. Additionally, we made sure to select ions which were still specific to the original molecular structure, instead of



reagent fragments such as *m/z* 181 for PFBHA. These non-specific reagent fragments can be dominant and therefore increase sensitivity, but are more prone to interferences by compounds with similar functional groups and the same or a similar retention time. For each compound, we selected the most abundant ion in the EI spectrum fulfilling these criteria. Additional information

on the peak systems and the measured ions is given in Table S4.

**Table 1: List of all compounds monitored by SPME-GC-MS.** Compound-specific abbreviations (Abb), retention times (RT), and molecular weights (MW) are given. The functional groups amenable to oxime formation (PFBHA derivatization) or silylation (MSTFA/TMCS derivatization) are denoted as aldehydes (CHO), ketones (C=O), alcohols (OH), and carboxylic acids (COOH). The selected
ion monitoring (SIM) masses refer to the derivatized compounds. Internal standards (ISTDs) were assigned based on the derivatization mechanism (acetophenone-$d_8$ for oxime formation, phenol-$d_6$ for silylation). Compounds are sorted according to their retention time. Dashed horizontal lines mark the time segments starting at 4.0, 9.8, and 15.0 minutes, respectively.

| Compound | Abb | RT / min | MW / g mol$^{-1}$ | Oxime formation | Silylation | SIM / m/z | ISTD |
|---|---|---|---|---|---|---|---|
| Phenol-$d_6$[a] | PHE-$d_6$ | 5.78 | 100.15 | - | $1 \times$ OH | 156.1 | - |
| o-Cresol | OCR | 7.01 | 108.14 | - | $1 \times$ OH | 165.1 | PHE-$d_6$ |
| m-Cresol | MCR | 7.15 | 108.14 | - | $1 \times$ OH | 165.1 | PHE-$d_6$ |
| p-Cresol | PCR | 7.28 | 108.14 | - | $1 \times$ OH | 165.1 | PHE-$d_6$ |
| Benzyl alcohol | BOH | 7.33 | 108.14 | - | $1 \times$ OH | 135.1 | PHE-$d_6$ |
| o-Nitrotoluene[b] | ONT | 7.50 | 137.14 | - | - | 91.1 | - |
| (Nitromethyl)benzene[b] | NMB | 7.96 | 137.14 | - | - | 91.1 | - |
| m-Nitrotoluene[b] | MNT | 8.05 | 137.14 | - | - | 137.1 | - |
| Benzoic acid | BAC | 8.20 | 122.12 | - | $1 \times$ COOH | 179.1 | PHE-$d_6$ |
| p-Methylcatechol | PMC | 10.42 | 124.13 | - | $2 \times$ OH | 268.1 | PHE-$d_6$ |
| Glycolaldehyde | GAL | 10.80 | 60.05 | $1 \times$ CHO | $1 \times$ OH | 312.1 | PHE-$d_6$ |
| Nitrocresols[c] | NCR | 11.61 | 153.14 | - | $1 \times$ OH | 210.1 | PHE-$d_6$ |
| Pyruvic acid | PAC | 11.75 | 88.06 | $1 \times$ C=O | $1 \times$ COOH | 340.1 | PHE-$d_6$ |
| Acetophenone-$d_8$[a] | APH-$d_8$ | 12.95 | 128.20 | $1 \times$ C=O | - | 323.1 | - |
| Benzaldehyde | BAL | 13.18 | 106.12 | $1 \times$ CHO | - | 301.1 | APH-$d_8$ |
| Glyoxal | GLY | 15.43 | 58.04 | $2 \times$ CHO | - | 448.1 | APH-$d_8$ |
| Methylglyoxal | MGL | 15.82 | 72.06 | $1 \times$ CHO, $1 \times$ C=O | - | 265.1 | APH-$d_8$ |
| p-Hydroxybenzaldehyde | PHB | 16.46 | 122.12 | $1 \times$ CHO | $1 \times$ OH | 389.1 | PHE-$d_6$ |

[a] *Internal Standard.*

[b] *Only NO$_2$ functional group.*

[c] *Co-elution of 2-nitro-p-cresol and 6-nitro-m-cresol.*

To account for losses in the transfer line, the extraction from the sampling cell, and the on-line ISTD addition, we calibrated the analytes determined by SPME-GC-MS directly from within the chamber. Joint stock solutions of the analytes were prepared in acetonitrile (ACN) at six different concentrations (c = 0 mM, 0.05 mM, 0.1 mM, 0.2 mM, 0.3 mM, 0.5 mM). For
each calibration level, 125 μL of the solution were injected into an installed round-bottomed flask and flushed into the chamber with a 5 SLPM N$_2$ flow. After 15 minutes, the flask was heated to 60 °C for 5 minutes to ensure complete vaporization. Once



the chamber was completely filled, three samples were measured by the on-line SPME-GC-MS system consecutively. The active sampling scheme during the calibration was exactly the same as during the experiments (Sect. 3.3.1). For the entire calibration, the ambient temperature was controlled at T = 298±1 K. In between the different calibration levels, we cleaned the

chamber, the introduction system, and the sampling system thoroughly. Each calibration level was preceded by a blank measurement of the chamber to account for remaining carry-over. With a total chamber volume of about 300 L, the calibration levels spanned between c = 0 – 12.3×10$^{10}$ molecules cm$^{-3}$, corresponding to mixing ratios of approximately 0 − 5 ppbv at standard pressure. The acquired responses of the analytes were corrected for the ISTD response, the blank value, and the determined wall loss rate. For each compound, a linear regression of these corrected relative responses against the

concentration in molecules cm$^{-3}$ was performed and forced through the origin. We calculated the limits of detection (LODs) as three times the standard deviation of the blank calibration sample. For most tested compounds, the LOD was in the low pptv range (median = 10.75 pptv for T = 298 K and standard pressure, Table 3 in Sect. 4.2.1).

For the chamber experiments, we calculated a total error of the concentration (*quantification error*) from the propagation of

the compound-specific instrumental error, the calibration error, and the experimental error. The instrumental error (5 % – 52 %, Table 3 in Sect. 4.2.1) was obtained as the mean of the relative standard deviations (RSDs) for each calibration level. It was primarily affected by the GC-MS response, the derivatization procedure, fibre effects, and peak integration. Furthermore, it reflects the degree to which the ISTD response and the analyte response co-varied. The calibration error (18 %) included the preparation and stability of the calibration solutions and their vaporization and transportation into the chamber. The

experimental error (10 %) resulted from variations in the flows and injections during chamber filling and was calculated from the variability of the monitored toluene start concentrations across all experiments.

### 3.2.2 PTR-ToF-MS for VOC and Product Screening with High Temporal Resolution

The PTR-ToF-MS used in this work (Ionicon, PTR-TOF 4000) has a mass resolution of >4000 m/Δm (full width at half maximum) for *m/z* > 147. It is equipped with a multichannel plate (MCP) detector and an internal permeation source of 1,3-

diiodobenzene (C$_6$H$_4$I$_2$) to calibrate the mass axis continuously. We operated the instrument with a drift tube pressure of 2.6 mbar, and a reduced electric field strength of E/N = 95 Td. For all PTR-ToF-MS data shown here, hydronium reagent ions (H$_3$O$^+$) were used as the primary ion (PI) in order to enable ionization with low fragmentation and to accommodate the analysis of diverse classes of products (Lindinger et al., 1998; Romano and Hanna, 2018). During the experiments, the chamber air was sampled continuously through a 1/8" PTFE line of about 5 m length with a flow of 50 sccm. Besides the analysis of toluene,

we focused here on the quantitative analysis of the major photooxidation products. All compounds were measured at their protonated mass (Table S5). For toluene, we monitored the $^{13}$C-isotope at *m/z* 94.0716 to avoid artefacts related to peak saturation. We averaged all data in the experiments and the calibrations by the minute.



All PTR-ToF-MS calibrations were performed from the chamber to better mimic the conditions and losses during the experimental measurements. For the photooxidation products, the general procedure was analogous to the SPME-GC-MS calibration, except for that the calibration solutions were prepared in water instead of ACN. The aqueous matrix was necessary to avoid solvent mixing ratios in the chamber in the ppmv range, but limited the range of studied compounds to water-solvable ones. We calibrated the cresols, benzyl alcohol, benzaldehyde, glyoxal, and methylglyoxal. The change in relative humidity during the calibrations compared to the experiments was negligible, as only 125 µL water were injected into the 300 L chamber. For each calibration level, the PI-normalized signal was averaged over a stable time frame of 15 minutes after equilibration. The calibration curves were derived from the blank-corrected values and forced through the origin. The LODs were calculated as three times the standard deviation of the response obtained during an instrumental blank measurement that was performed with an activated charcoal filter. Except for glyoxal (LOD = 1827.8 pptv), the LODs varied between 0.5 pptv and 4.8 pptv (Table S6). Same as for the SPME-GC-MS, we derived the instrumental error as the mean RSD of all calibration levels, and calculated the quantification error from the propagation of the instrumental error (1 % – 2 % except for glyoxal with 89 %, Table S6), the calibration error (18 %), and the experimental error (10 %).

For toluene, we evaluated the measured signals in the filled chamber after equilibration and prior to the ignition of the solar simulator across all experiments (Sect 3.3). These experiments were carried out over a period of 6 weeks, during which time the sensitivity of the instrument is not susceptible to substantial drift. We calibrated toluene using the mean signal for its calculated start concentration of $c = 3.79 \times 10^{12}$ molecules cm$^{-3}$ and the blank value of the cleaned chamber. The overall quantification error was 10 %, calculated as propagation of the instrumental error (0.30 %) and the experimental error (10 %, variability of monitored start concentrations).

To highlight the benefit of the combined analytical instrumentation, we aimed to resolve the $C_7H_8O$ isomers also in the PTR-ToF-MS signal. Therefore, we performed separate calibrations for each of the three cresol isomers and benzyl alcohol. These calibrations were conducted on-line from the BATCH Teflon chamber analogously to the joint PTR-ToF-MS calibration. All compounds were analysed at the protonated mass of $m/z$ 109.0626, yet the instrumental response of each of the isomers was derived individually. For analyzing the sum signal during the photooxidation experiments, we calculated the weighted sensitivity for each of the isomers as the product of its instrumental sensitivity and its relative abundance. The relative abundances were derived as fixed values from the formation yields as determined with the SPME-GC-MS data (Sect. 4.3.2). The sum of these weighted sensitivities was used as the weighted calibration factor for the $m/z$ 109.0626 signal (Table S6). From the correctly quantified sum signal, we extracted the concentrations of the individual $C_7H_8O$ isomers using the same fixed relative abundances.





## 3.3 Photooxidation Experiments and Data Analysis


In total, we performed 18 experiments at different temperatures and with different initial $NO_x$ mixing ratios for method development purposes. To evaluate the product formation yields and gain mechanistic insight into the toluene chemistry, we focused here exclusively on six $NO_x$-free toluene-OH photooxidation experiments at T = 298±1 K and different degrees of photochemical aging (Table 2). We decided to work with an excess of toluene to keep secondary chemistry to a minimum and

produce sufficient product mass for analysis.

**Table 2: List of conducted $NO_x$-free experiments at T = 298±1 K.** The monitored start concentration of toluene spread around the calculated start concentration of c = $3.79\times10^{12}$ molecules cm$^{-3}$. The timing of the first (and all consecutive) SPME-GC-MS samples was varied to better constrain formation yields.

| Label | Toluene / molecules cm$^{-3}$ | Lights / min | Sample 1 / min |
|---|---|---|---|
| Tol-OH-1 | $3.40\times10^{12}$ | 90 | 20 |
| Tol-OH-2 | $3.23\times10^{12}$ | 75 | 10 |
| Tol-OH-3 | $4.18\times10^{12}$ | 30 | 20 |
| Tol-OH-4 | $3.93\times10^{12}$ | 45 | 30 |
| Tol-OH-5 | $4.12\times10^{12}$ | 60 | 40 |
| Tol-OH-6 | $3.79\times10^{12}$ | 105 | 10 |
| Control | - | 60 | 40 |


### 3.3.1 Experimental Protocol

Prior to each experiment, we cleaned the chamber and recorded blank values. To dilute and remove remaining impurities, the chamber was filled with zero air and then evacuated three times in total. Zero air was prepared by purifying pressurized synthetic air with a commercial zero air generator (Messer Griesheim GmbH, SL 50). In the first flushing cycle, the solar

simulator was ignited to promote the oxidation of potential residuals and their release from the walls. After the cleaning, preliminary chamber blanks were measured by the SPME-GC-MS and the PTR-ToF-MS to confirm the cleanliness of the chamber and to test the instrumental performances. For the SPME-GC-MS, this concerned in particular the condition of the SPME fibre as indicated by the PFBHA and ISTD responses. For the PTR-ToF-MS, the background concentration of toluene was obtained. After these checks, the chamber was completely evacuated in preparation for the experiment. The temperature

was set to the target value.

Upon completion of the preparatory work, we added the reagents sequentially. First, we introduced the VOC precursor into the chamber. The pure compound was injected through a septum into a 100 mL round-bottomed flask, which was flushed with $N_2$ as carrier gas (5 SLPM) for 15 minutes. In this work, we used 0.2 μL toluene (Roth, ≥ 99.9 % purity), equivalent to a target

concentration of $3.79\times10^{12}$ molecules cm$^{-3}$ (154 ppbv at T = 298 K and standard pressure). Second, hydrogen peroxide ($H_2O_2$)



was added as OH radical precursor by filling 3 mL of a 50 % aqueous solution (Sigma Aldrich) into a 5 mL impinger to enrich a flow of $N_2$ (100 sccm) which was entering the chamber for 20 minutes. We selected $H_2O_2$ because it provides a $NO_x$-free method of generating OH radicals and photolyzes even under the near-UV conditions in the chamber. If needed, NO was then introduced directly from a gas cylinder (Rießner-Gase GmbH, 2.88 ppmv in $N_2$). Finally, the chamber was filled to 300 L with

zero air (10 SLPM). All flows were regulated by mass flow controllers and determined with flow meters.

After complete filling, the SPME-GC-MS, the PTR-ToF-MS, and if needed the $NO_x$ and $O_3$ analysers, were connected to the chamber. The true experimental blank of the oxidation products was obtained in the presence of toluene and $H_2O_2$, and the start concentration of toluene was monitored. Afterwards, the solar simulator was turned on to initiate the photooxidation

chemistry. The lights were kept covered for 3 minutes after ignition, as the emission of the lamps during this warm-up period is generally small and not reproducible (Bleicher, 2012; Wittmer et al., 2015). The start of the oxidation process was marked by the uncovering of the lights. For each SPME-GC-MS sample, the vacuum pump for active sampling was started 3 minutes prior to the on-line extraction and stopped again afterwards in order to conserve chamber volume but still allow sufficient time for equilibration. Since all instruments were attached to the same outlet line of the chamber, the PTR-ToF-MS sample flow

(50 sccm) was affected by the active sampling of the SPME-GC-MS pump (5 SLPM) every 41 minutes. This effect was minimized by attaching the PTR-ToF-MS transfer line as closely as possible to the chamber (~5 cm distance from the chamber interior). Remaining artefacts in the PTR-ToF-MS signal were corrected by linear interpolation using the data acquired 2 min before and after the SPME-GC-MS pump was turned on, respectively.

The chamber was never depleted to less than 30 % of its initial volume to avoid too high surface-area-to-volume ratios and an increased importance of wall effects. After all the data were collected, the chamber was evacuated and then cleaned three times (zero air filling and subsequent evacuation). In the final cycle, the chamber was evacuated by the SPME-GC-MS pump to also clean the transfer line and sampling cell.

For each experiment, air pressure data were retrieved from the weather station at the Botanical Garden of the University of Bayreuth (4 km distance from the atmospheric chemistry laboratory) and averaged over the experimental time frame. We measured the relative humidity in the chamber in one of the experiments and derived a value of 0.1 %. This is consistent with the composition of the air mixture, since the only source of water in the chamber was the $N_2$ flow enriched with the $H_2O_2/H_2O$ solution, accounting for only 2 L of the 300 L total chamber volume in all performed experiments.

**3.3.2 Initial Data Treatment**

The raw GC-MS data were processed using MassHunter (GC/MS Data Acquisition v10.1.49, Qualitative Analysis v10.0, Quantitative Analysis v10.2) for peak integration. All further data analysis was performed in Python (v3.9.7). After normalizing the responses to the assigned ISTDs (relative responses), they were blank-subtracted and converted to





concentrations in molecules cm$^{-3}$. For the PTR-ToF-MS data, we used the IONICON Data Analyzer (v2.1.1.2) for mass axis

calibration, peak definition, and PI-normalization. In Python, we compiled mean values per minute, subtracted blank values, and applied the calibration factors.

### 3.3.3 Loss Corrections

Corrections for compound-specific physical and chemical losses were needed to obtain the true formation yields of the studied photooxidation products. In the present work, these losses included wall losses, photolysis, and secondary OH radical reactions.

As we evaluated the yields for NO$_x$-free conditions, secondary reactions with O$_3$ and NO$_3$ radicals as competing oxidants were irrelevant. We corrected the losses for each discrete data point iteratively in order to account for the simultaneous formation and loss of the photooxidation products. The calculation is shown in Eq. (2):

$$\Delta L(\Delta t) = \bar{c}(\Delta t) \times \Delta t \times k \tag{2a}$$

$$c_t^{corr} = c_{t-1}^{corr} + \Delta c(\Delta t) + \Delta L(\Delta t) \tag{2b}$$

where $\Delta L(\Delta t)$ is the absolute loss in units of molecules cm$^{-3}$ that occurred over the time span between two given data points. This loss is defined by $\overline{c}(\Delta t)$ as the mean value of the measured concentrations $c_t$ and $c_{t-1}$ in molecules cm$^{-3}$, $\Delta t$ as the time span between the two data points in s, and $k$ as the first order loss rate constant for the specific loss process in s$^{-1}$. The total product concentration formed up to a given point in time, *loss-corrected concentration* hereinafter, entails both the measured concentration and the cumulative loss that occurred since the beginning of the experiment (Fig. S8). The loss-corrected

concentration of a given data point $c_t^{corr}$ is therefore calculated as the sum of the loss-corrected concentration of the previous data point $c_{t-1}^{corr}$, the absolute loss that occurred since that point in time $\Delta L(\Delta t)$, and $\Delta c(\Delta t)$ as the measured difference between these two data points (all in molecules cm$^{-3}$).

This procedure is based on the approach described by Galloway et al. (2011), but was adapted for the low temporal resolution

of the SPME-GC-MS method by relating the loss term to $\overline{c}(\Delta t)$ rather than to the measured concentration of the previous data point. The calculation was performed for each of the three loss processes individually, so as to better account for the conditions in the chamber, e.g. light or dark, and in order to derive the absolute losses associated with each process. All three corrections were implemented with reference to the measured data and then summed up to derive the final loss-corrected concentration (Fig. S8). The calculations were performed for the SPME-GC-MS and the PTR-ToF-MS analogously. In cases where different

compounds with an unknown distribution contributed to a sum signal at the PTR-ToF-MS, we selected the rate constants of the compound which we assumed to be most abundant.

Both the wall loss rates and the photolysis rates were available as first order rate constants. While wall losses were corrected for the entire experimental duration, photolytic losses were corrected only for the periods in which the solar simulator was

ignited. To treat the reaction with OH radicals using the same correction formula, we converted the second order reaction of any of the analytes with OH radicals to a pseudo first order approach, as shown in Eq. (3):



$$\Delta t \times k' = \Delta t \times k \times [OH] = \Delta OHexp \times k \qquad (3)$$

where $k'$ is the pseudo first order rate constant in s$^{-1}$, $k$ is the second order rate constant in molecules$^{-1}$ cm$^3$ s$^{-1}$, $[OH]$ is the OH radical concentration in molecules cm$^{-3}$, and $\Delta OHexp$ is the difference of the OH exposure between the two data points which is given in the units of molecules cm$^{-3}$ s. The second order rate constants for the reactions of the photooxidation products with OH radicals were obtained from the IUPAC recommendations and the scientific literature (Table S7). The calculation of the OH exposure at a given point in time was based on the observed decay of toluene to circumvent the need for direct OH radical measurements, as shown in Eq. (4):

$$\Delta OHexp = OHexp_t - OHexp_{t-1} \qquad (4a)$$

$$OHexp_t = \ln \left( \frac{[toluene]_t}{[toluene]_0} \right) / -k_{toluene+OH} \qquad (4b)$$

where $OHexp_t$ and $OHexp_{t-1}$ are the OH exposures determined for the time steps associated with the target data point and the previous data point in molecules cm$^{-3}$ s, $[toluene]_t$ and $[toluene]_0$ are the concentrations of toluene at the given point in time and at the beginning of the experiment in molecules cm$^{-3}$, and $k_{toluene+OH} = 5.6 \times 10^{-12}$ molecules$^{-1}$ cm$^3$ s$^{-1}$ is the well-characterized second order rate constant of toluene with OH radicals (IUPAC, 2024).

### 3.3.4 Yield Calculations

The formation yields of the first generation products that we observed, identified, and quantified in the new chamber for T = 298±1 K in the absence of NO$_x$ were calculated from a large data set assembled from six experiments (Table 2). For each compound, we plotted the loss-corrected concentration against the change of toluene for a given point in time, performed a linear regression of the collected data set, and determined the yield as the slope of this regression. This approach helps to minimize the impact of the random error by regressing over a broad range of precursor losses (e.g. Atkinson et al., 1989; Olariu et al., 2002). We included only those data for which the non-corrected value was higher than the LOD, to enhance accuracy and reliability. Since the data acquired at the end of the experiments were affected to a higher extent by secondary chemistry and chamber effects, we considered only the data obtained while the chamber was irradiated as well as the first sample in the dark in case of the SPME-GC-MS.

## 4 Results and Discussion

An exemplary time series of toluene and its ring-retaining first generation products for one of the experiments at T = 298±1 K under NO$_x$-free conditions is shown in Fig. 3. While the chamber was irradiated, OH radicals were generated and caused a decrease in toluene and the formation of products. No loss of toluene was observed after the lights were turned off, indicating the termination of the oxidation chemistry in the absence of photolytic OH radical generation.



Under the studied conditions, the OH radical reaction was the only relevant sink of toluene, so that the change in its concentration could be directly used for the calculation of OH radical concentrations and product formation yields. The photolysis rate of toluene in the irradiated chamber was calculated to be as low as $J = 3.78\pm1.44\times10^{-7}$ s$^{-1}$ (Table S1). Due to the high vapor pressure (21.78 torr at T = 293 K (Munday et al., 1980)) and the low degree of functionalization of toluene, losses to the chamber wall were not substantial. Its rate constants for reactions with O$_3$ and NO$_3$ radicals are about k $<10^{-21}$ molecules$^{-1}$ cm$^3$ s$^{-1}$ and k $= 7.8\times10^{-17}$ molecules$^{-1}$ cm$^3$ s$^{-1}$ (IUPAC, 2024), so that these loss processes were irrelevant under all experimental conditions.

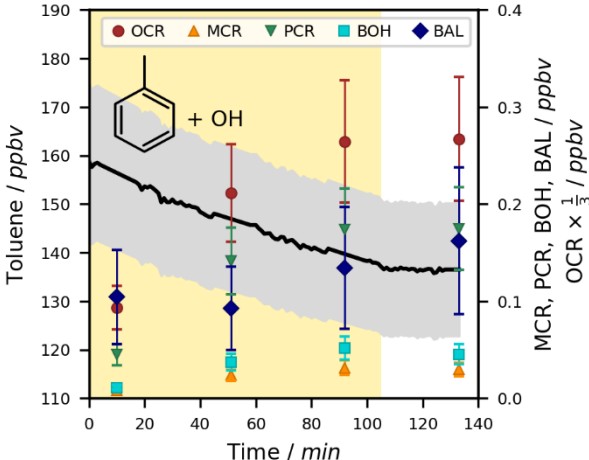

**Figure 3: Exemplary time series for the OH radical-induced photooxidation of toluene at T = 298±1 K in the absence of NO$_x$.** The temporal profiles of toluene and its ring-retaining first generation products in experiment Tol-OH-6 are shown. Toluene was determined by PTR-ToF-MS (black line, quantification error shown as grey area), while the products were analysed using the on-line double-derivatization SPME-GC-MS method (quantification error shown as error bars). No loss corrections were applied to the depicted data. The area shaded in yellow represents the period in which the chamber was irradiated.

## 4.1 Chamber Characterization

The new BATCH Teflon chamber was built for the purpose of studying multifunctional gas-phase products. Losses and uncertainties in the quantification of these target species result amongst others from the interaction with the chamber setup. Thus, we first characterized the spectrum of the solar simulator. Secondly, we derived typical wall losses of the photooxidation products and proposed a chamber-specific parameterization. Thirdly, we assessed possible carry-over and artefacts.

### 4.1.1 Solar Simulator Spectrum and Photolysis Rates

The measured emission spectrum of the solar simulator is shown in Fig. 4. On average, the emission intensity in our recorded spectrum was about 3 times higher compared to the previous measurement by Bleicher (2012), which was obtained by



differential optical absorption spectroscopy (DOAS) and normalized to the empirical $Cl_2$ photolysis rate. This difference could relate to the specific emission and age of each lamp, or to the vertical distance at which the previous spectrum was recorded.

Using the newly measured spectrum ($\lambda = 325 - 1000$ nm) and the spectrum scaled from the old data set ($\lambda = 262 - 325$ nm), the $NO_2$ photolysis rates calculated theoretically ($2.07\pm0.79\times10^{-2}$ s$^{-1}$) and derived experimentally ($2.32\pm1.99\times10^{-2}$ s$^{-1}$, Fig. S2) agree within 12 %. For methylglyoxal, the experimental photolysis rate ($3.66\pm0.56\times10^{-4}$ s$^{-1}$, Fig. S3) is 1.21-fold higher than the calculated value ($3.02\pm1.15\times10^{-4}$ s$^{-1}$), yet the uncertainties of these rates overlap.

By design, the HMI lamp housings filter radiation $\lambda < 260$ nm (Siekmann, 2018) in order to block for instance the 185 nm and 254 nm mercury emission lines. By shifting the UV absorption edge to about $\lambda < 300$ nm, the bandpass filter of the solar simulator reproduced a more realistic photon flux in the UV range. For comparison, Fig. 4 shows the resemblance between the solar simulator emission spectrum and the natural solar spectrum, as calculated for a latitude of 49°N with the National Center for Atmospheric Research (NCAR) Tropospheric Ultraviolet and Visible (TUV) radiation model v5.3 (NCAR, 2024) for 510 midday in June and December.

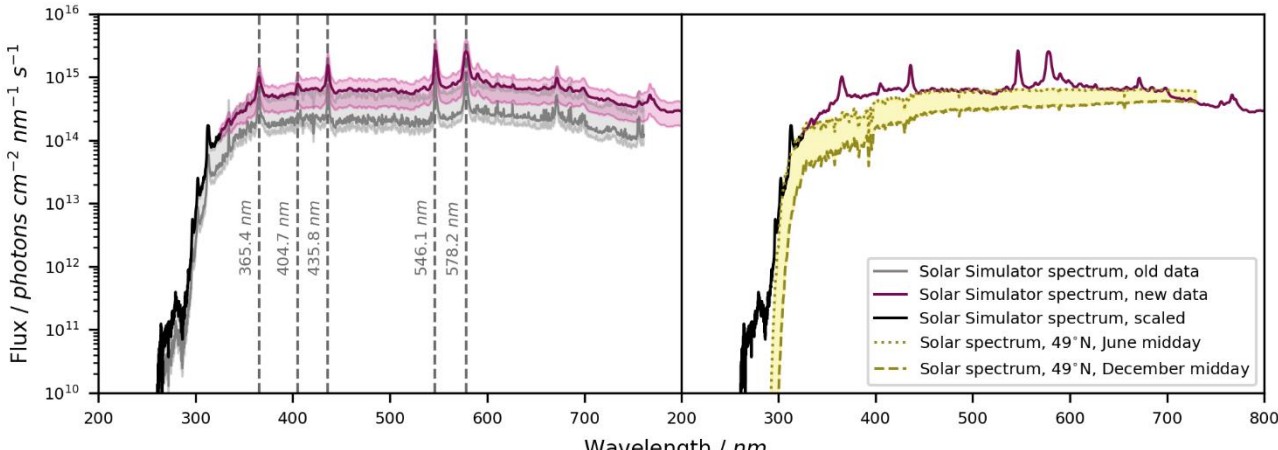

**Figure 4: Solar simulator emission spectrum.** The left figure shows the emission spectrum of the solar simulator (7 HMI lamps with bandpass filter) available from an old measurement (Bleicher, 2012), the newly recorded spectrum, and the scaled spectrum in the wavelength range between $\lambda = 262 - 325$ nm (scaled by comparing the new data and the old data for the wavelength range $\lambda = 350 - 500$ nm). The new 515 spectrum is shown as the mean ± standard deviation from 30 measurements at different positions above the solar simulator. The uncertainty in the old data results from the quantification of Cl radicals used to calibrate the actinic flux. The main Hg emission lines are shown. In the right figure, the solar simulator spectrum is compared against the natural actinic flux as calculated with the TUV radiation model for 49°N for midday in June and December respectively (NCAR, 2024). Figure concept inspired by Bleicher (2012).

The emission of the solar simulator directly affected the photolysis of $H_2O_2$ and thereby the OH radical source. The photolysis of $H_2O_2$ generates OH radicals with a near-unity quantum yield (Vaghjiani and Ravishankara, 1990) and occurred with a rate of about J = $1.13\pm0.43\times10^{-5}$ s$^{-1}$ in the irradiated chamber (Fig. S1, Table S1). For $NO_x$-free experiments at T = $298\pm1$ K, the



resulting mean OH radical concentration was [OH] = $4.89 \times 10^6$ molecules cm$^{-3}$ while the solar simulator was activated. This is comparable to tropospheric daytime OH radical levels, which typically peak in the range of $2 \times 10^6 - 8 \times 10^6$ molecules cm$^{-3}$ for

similar latitudes (Altshuller, 1989). We did not observe any product formation in a test run without $H_2O_2$ addition, confirming $H_2O_2$ photolysis as the main source of OH radicals, rather than for example the release of HONO from chamber walls which would produce both OH radicals and NO.

The calculated photolysis rates of the studied oxidation products (Table S1) were in the range between $1.77 \pm 0.67 \times 10^{-8}$ s$^{-1}$ (*m*-

nitrotoluene) and $3.02 \pm 1.15 \times 10^{-4}$ s$^{-1}$ (methylglyoxal). High photolysis rates of $>10^{-4}$ s$^{-1}$ were determined for the carbonyl compounds benzaldehyde, glyoxal, methylglyoxal, and pyruvic acid. Specific quantum yields were mostly available for those compounds which are prone to absorb in the solar actinic spectrum and have therefore been extensively investigated with regard to their photochemistry, such as carbonyls and nitro compounds. For the studied alcohols and carboxylic acids for which no quantum yield was available in the literature, we determined maximum photolysis rates of $<2 \times 10^{-5}$ s$^{-1}$ with assumed

quantum yields of 1. These loss rates do not substantially affect the chemistry and calculated concentrations over the course of the experiment, so that over-corrections of the photolytic losses are negligible.

### 4.1.2 Wall Losses

The wall loss rates which were determined empirically in this work (Table S8) ranged between $4.54 \pm 1.80 \times 10^{-6}$ s$^{-1}$ (benzaldehyde) and $8.53 \pm 0.68 \times 10^{-5}$ s$^{-1}$ (*p*-hydroxybenzaldehyde). These orders of magnitude are in line with reported vapor

loss rates on Teflon films in the literature (McMurry and Grosjean, 1985; Zhang et al., 2015).

The dependence of the measured wall loss rates of the studied oxygenated compounds on their molecular properties is shown in Fig. 5. The most influential parameter was the vapor pressure, which has already been shown previously to be inversely related to wall losses in Teflon chambers (Yeh and Ziemann, 2015; Zhang et al., 2015). In addition, we found a dependency

on the molecular weight of the compounds, which is similar to other studies where partitioning to walls increased with the carbon number of the molecule (Matsunaga and Ziemann, 2010; Yeh and Ziemann, 2015). Finally, the wall loss was also influenced by the degree of functionalization and oxygenation (Lumiaro et al., 2021; Matsunaga and Ziemann, 2010; Yeh and Ziemann, 2015), considered in this study in terms of the oxygen-to-carbon ratio (O:C ratio). Here, we propose a parameterization for the wall loss rate ($R^2 = 0.80$) as given in Eq. (5):

$$WL = a \times e^{(b \times P)} \tag{5a}$$

$$P = \frac{\sqrt{v_p}}{MW \times OC} \tag{5b}$$

where $WL$ is the wall loss rate in s$^{-1}$, $a$ is $6.49 \times 10^{-5}$ s$^{-1}$, $b$ is -19.68 ($\sqrt{\text{mmHg}}$)$^{-1}$ g mol$^{-1}$, and $P$ is the parameter in $\sqrt{\text{mmHg}}$ g$^{-1}$ mol. The parameter $P$ entails the vapor pressure $v_p$ in mmHg, the molecular weight $MW$ in g mol$^{-1}$, and the dimensionless O:C ratio $OC$. We observed a different trend for nitro compounds without further functional groups, and did not include their





empirical wall loss rates for fitting the regression. The empirical and parameterized wall loss rates and their relative deviations are summarized for all oxygenated products in Table S8. With the exception of benzaldehyde and methylglyoxal, which both have small empirical loss rates with high uncertainties ($<2\times10^{-5}$ s$^{-1}$, uncertainty of 40 % and 32 % respectively), the empirical and parameterized values agree within 30 %.

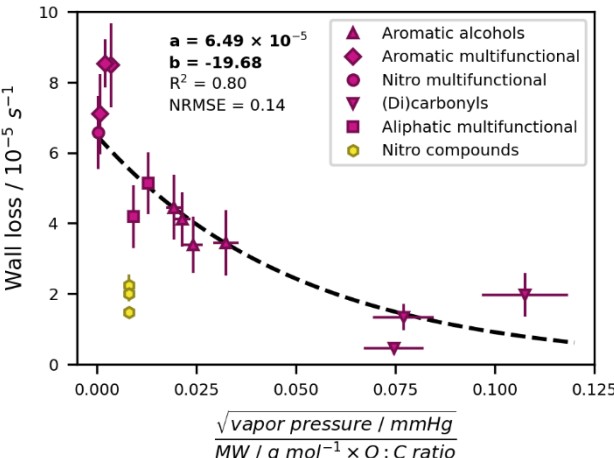


**Figure 5: Wall loss parameterization based on empirically determined wall loss rates.** Error bars along the y-axis are the normalized root mean squared errors (NRMSEs) of the individual exponential regressions for the empirical wall loss determination. Error bars along the x-axis show a generic 10 % error resulting from uncertainties in available vapor pressure data. Oxidation products with different functional groups are distinguished. Nitro compounds (nitrotoluene isomers and (nitromethyl)benzene) were not included as model input.


In a larger context, the presented parameterization provides the possibility to describe the chamber-specific wall loss of a multitude of oxygenated photooxidation products which cannot be tested empirically. For higher internal consistency, we used the parameterized loss rates for all studied compounds with hydroxy, carboxyl, or carbonyl groups. These values are associated with an uncertainty of 14 % according to the normalized root mean square error (NRMSE) of the parameterization.


The application of the parameterization is limited in that reasonably well-constrained vapor pressure is required to make an explicit prediction. In the absence of such data, the loss rate can be approximated by generic values for clusters of compounds with the same functional group composition. Here, six groups of compounds could be clearly distinguished (see also Table S8). Using the fitted wall loss rate of all associated compounds, we derived generic loss rates (mean ± standard deviation) of

$1.23\pm0.32\times10^{-5}$ s$^{-1}$ for carbonyls and dicarbonyls (benzaldehyde, glyoxal, methylglyoxal), $4.04\pm0.38\times10^{-5}$ s$^{-1}$ for monohydric aromatic alcohols (cresols and benzyl alcohol), $5.23\pm0.19\times10^{-5}$ s$^{-1}$ for aliphatic multifunctional compounds (glycolaldehyde, pyruvic acid), $6.23\pm0.14\times10^{-5}$ s$^{-1}$ for aromatic acids and multifunctional compounds (benzoic acid, p-methylcatechol, p-hydroxybenzaldehyde), and $6.44\times10^{-5}$ s$^{-1}$ for nitro compounds with additional hydroxy groups (nitrocresol isomers). The order of the associated wall loss rates (multifunctional compounds and acids > alcohols > carbonyls) agrees well with previous



studies which have investigated the effect of the type (Yeh and Ziemann, 2015) and number (Lumiaro et al., 2021) of functional groups on partitioning coefficients of gaseous compounds to walls and particles. Although not included in the parameterization, nitro compounds can be described by their mean empirical wall loss rate of $1.90\pm0.32\times10^{-5}$ s$^{-1}$.

### 4.1.3 Cleanliness and Quality Control

To minimize carry-over effects and quantify chamber contamination, the cleanliness of the chamber was examined carefully
prior to each experiment. During these checks from the pre-cleaned chamber, the product concentrations measured in the irradiated chamber and in the dark chamber agreed well within the instrumental error. Hence, there was no substantial desorption of residual compounds from the irradiated walls. In addition, we measured the experimental blank values from the filled chamber to prevent overestimations of product concentrations due to impurities and analytical artefacts related to the added reagents. For most compounds, these blank values were in a similar range as their respective LODs, by several orders
of magnitude smaller than their typical concentrations during the photooxidation experiments, and did not differ substantially among the different runs (Fig. 6).

To confirm that the observed product formation during the photooxidation experiments is solely attributable to the reaction of toluene with OH radicals, we conducted an additional control experiment in the absence of toluene (Table 2). The
concentrations obtained in the course of this experiment are comparable to the measured blank values (Fig. 6). Hence, we conclude that the reacted toluene was the unique source of the products in our setup.

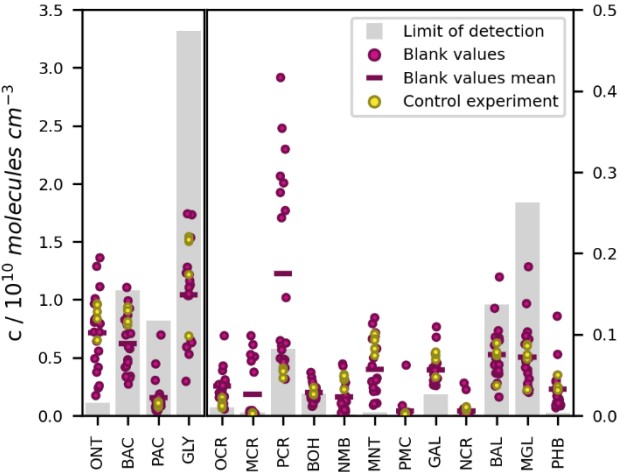

**Figure 6: LODs, blank values, and control experiment concentrations of the toluene photooxidation products measured by SPME-GC-MS.** The LODs refer to the on-line measurement from the chamber. Blank values are shown for 18 separate experiments. The measured
concentrations in the toluene-free control experiment are shown without blank subtraction or further loss corrections. All values are given as concentrations in molecules cm$^{-3}$, because the LODs for the SPME-GC-MS were derived as fixed concentrations, whereas the corresponding mixing ratios varied between the experiments due to the ambient temperature and pressure.



**4.2 Analytical Performance Evaluation: SPME-GC-MS and PTR-ToF-MS**

While it was possible to quantify the isomers of functionally diverse photooxidation products at ppbv and pptv levels with the
double derivatization SPME-GC-MS method, its temporal resolution was limited to 41 minutes. In contrast, PTR-ToF-MS
averaged scans were monitored every minute with high precision and additional information about potential photooxidation
products yet no possibility to distinguish between isomers. Thus, here we combined the two analytical tools to extend and
improve the information derived from the conducted photooxidation experiments, merging isomeric information from the
SPME-GC-MS and a high time resolution from the PTR-ToF-MS.

**4.2.1 SPME-GC-MS Analysis and Calibration of Toluene Products**

For the compounds which could be analysed efficiently with the on-line SPME-GC-MS method, the determined calibration
factors, associated errors, and LODs are listed in Table 3. The calibration curves are shown exemplary for the ring-retaining
first generation products in Fig. 7. No saturation effect is apparent in these curves, indicating that the calibration interval
between $c = 0 - 12.3 \times 10^{10}$ molecules cm$^{-3}$ is still within the linear range of the SPME fibre. We observed the lowest sensitivity
for compounds undergoing both derivatization steps (glycolaldehyde, pyruvic acid, *p*-hydroxybenzaldehyde) and the highest
instrumental sensitivity for the non-derivatized nitrotoluene isomers and (nitromethyl)benzene. This is partly due to the
selection of ions for the SIM mode and the relatively more complex mass spectrum after derivatization. While non-derivatized
compounds were monitored on predominant ions in their mass spectra, the SIM masses of the derivatized compounds were
chosen more specifically to represent typical adducts with all involved derivatization reagents.


The instrumental error was highest for the carbonyls benzaldehyde, glyoxal, and methylglyoxal that were derivatized by
PFBHA and corrected by acetophenone-d$_8$. Along with the two carboxylic acids, benzoic acid and pyruvic acid, these
compounds furthermore had the highest LODs. This is in contrast to the study by Borrás et al. (2021), where carbonyls could
be determined with great precision, and where especially glyoxal performed very well. Probably, this difference results from
the distinct experimental setups of the BATCH Teflon chamber and the EUPHORE chamber, respectively. In particular, the
transfer line in the present work was longer (~ 6 m), heated to 50°C instead of 80 °C, and made from PTFE instead of sulfinert
material. Additionally, the periodic sampling process allowed only 3 minutes of equilibration prior to extraction. We
acknowledge that these conditions limit our ability to analyse sticky compounds with regard to sensitivity and sample-to-
sample variability (Fig. S9). Furthermore, during preliminary stability and optimization tests, we sometimes observed declining
responses of PFBHA and the associated analytes over time, either due to reagent depletion or fibre degradation (Fig. S10). We
avoided such effects in the experimental data by checking the fibre condition prior to each experiment and by conducting
relatively short experiments with not more than five samples. Still, we note that the derivatization with PFBHA may be less
robust than the derivatization with MSTFA/TMCS. Although such effects should be accounted for by the ISTD correction,





acetophenone-d$_8$ as the selected ISTD may be too structurally different from the photooxidation products with regard to

aromaticity and the type of carbonyl group to be able to sufficiently correct for pronounced methodological variations.

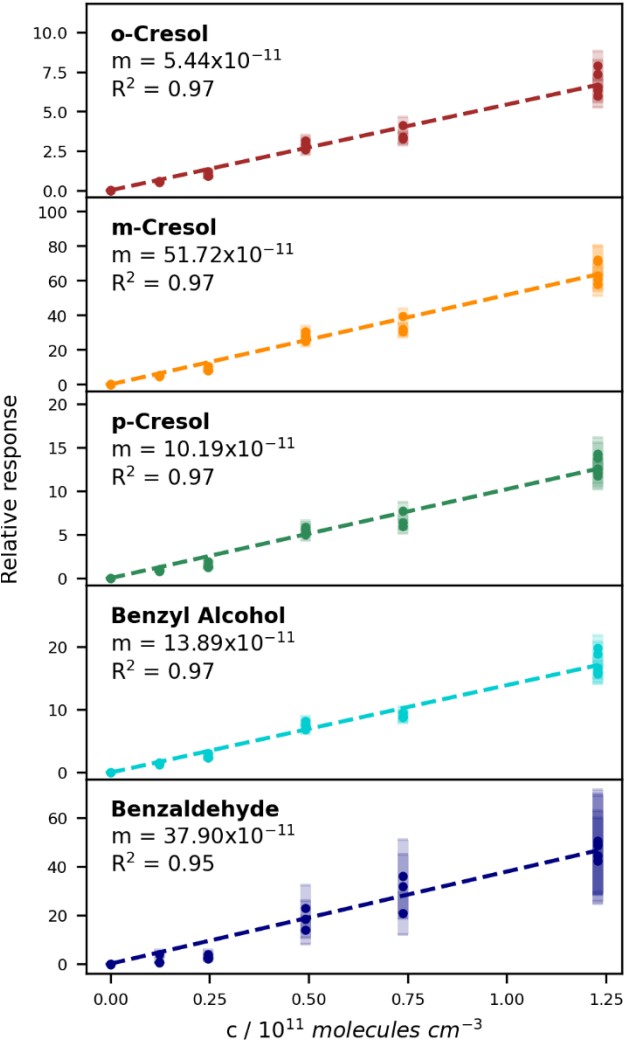

**Figure 7: Calibration curves for the ring-retaining first generation products of toluene with the ISTD correction.** The relative responses are plotted against the concentration in the chamber. Uncertainty areas represent the instrumental error.

The calibration solutions were prepared in ACN because (i) all compounds were solvable at the target concentrations, and (ii) ACN contains no functional groups that could scavenge the derivatization reagents when present in excess. However, this choice of solvent led to a substantial difference of the chamber matrix between the calibration and the experiments. In the present study, any potential matrix-induced changes in the instrumental sensitivity were compensated for by the on-line ISTD correction. Additionally, matrix-specific blank samples were subtracted both in the calibration (injection of ACN into chamber



without dissolved analytes) and the experiments (filled chamber prior to irradiation) to account for possible variations in instrumental offsets. To experimentally confirm the validity of the performed calibration, joint stock solutions of the cresol isomers, benzyl alcohol, benzaldehyde, glyoxal, and methylglyoxal were prepared in ACN and water respectively (c = 0.5 mM). For each solvent, 125 µL of the stock solution were injected into the chamber and measured by SPME-GC-MS. For all tested compounds, the uncertainty ranges of the mean relative responses of the two solutions overlapped. Hence, any potential

solvent effect was within the instrumental error, and artefacts related to the presence of ACN can be excluded.

**Table 3: Calibration factors, errors, and LODs for the compounds analysed with the on-line SPME-GC-MS method.** The uncertainty of the slope was derived from the standard error of the slope with a 95 % confidence interval. $R^2$ is the coefficient of determination of the regression performed for the calibration. The instrumental error (Instr. Err.) is the mean RSD of all calibration levels. The quantification
error (Quant. Err.) includes the instrumental error, the calibration error, and the experimental error. The LODs for the SPME-GC-MS were derived as fixed concentrations in molecules cm$^{-3}$, whereas the corresponding mixing ratios varied between the experiments due to the ambient conditions. As information for the reader, the LOD in pptv is shown here for T = 298 K and standard pressure.

| Compound | Calibration | | Relative Error | | Limit of Detection (LOD) | |
| --- | --- | --- | --- | --- | --- | --- |
| | Slope / molecules$^{-1}$ cm$^3$ | $R^2$ | Instr. Err. / % | Quant. Err. / % | LOD / molecules cm$^{-3}$ | LOD / pptv |
| OCR | $5.44\pm0.47\times10^{-11}$ | 0.97 | 12 | 24 | $1.06\times10^8$ | 4.3 |
| MCR | $5.17\pm0.43\times10^{-10}$ | 0.97 | 12 | 24 | $3.49\times10^7$ | 1.4 |
| PCR | $1.02\pm0.08\times10^{-10}$ | 0.97 | 13 | 24 | $8.21\times10^8$ | 33.4 |
| BOH | $1.39\pm0.11\times10^{-10}$ | 0.97 | 10 | 23 | $2.66\times10^8$ | 10.8 |
| ONT | $6.94\pm0.53\times10^{-7}$ | 0.98 | 7 | 22 | $1.11\times10^9$ | 45.1 |
| NMB | $1.64\pm0.07\times10^{-6}$ | 0.99 | 5 | 21 | $1.45\times10^7$ | 0.6 |
| MNT | $9.05\pm0.43\times10^{-7}$ | 0.99 | 5 | 21 | $4.14\times10^7$ | 1.7 |
| BAC | $8.66\pm1.22\times10^{-11}$ | 0.93 | 31 | 37 | $1.08\times10^{10}$ | 438.9 |
| PMC | $4.44\pm0.64\times10^{-11}$ | 0.91 | 15 | 25 | $8.94\times10^6$ | 0.4 |
| GAL | $6.55\pm0.82\times10^{-11}$ | 0.94 | 12 | 24 | $2.63\times10^8$ | 10.7 |
| NCR[a] | $3.01\pm0.39\times10^{-10}$ | 0.94 | 14 | 25 | $1.42\times10^7$ | 0.6 |
| PAC | $8.81\pm1.38\times10^{-12}$ | 0.91 | 16 | 26 | $8.20\times10^9$ | 333.2 |
| BAL | $3.79\pm0.44\times10^{-10}$ | 0.95 | 42 | 47 | $1.37\times10^9$ | 55.6 |
| GLY | $2.02\pm1.05\times10^{-11}$ | 0.34 | 52 | 56 | $3.32\times10^{10}$ | 1349.5 |
| MGL | $3.59\pm0.47\times10^{-10}$ | 0.94 | 42 | 47 | $2.63\times10^9$ | 106.8 |
| PHB | $4.72\pm0.73\times10^{-12}$ | 0.92 | 26 | 33 | $3.26\times10^7$ | 1.3 |

[a] *Evaluated as the average concentration of both 2-nitro-p-cresol and 6-nitro-m-cresol.*

Optimizing and testing the new SPME-GC-MS method, we also evaluated several compounds which we did not include for the characterization of the BATCH Teflon chamber due to insufficient sensitivity or poor transportation from the chamber to the instrument (Table S9). We observed some patterns that can be of general interest for method optimization of these compounds, as described in more detail in the Supplement S6.3.



## 4.2.2 SPME-GC-MS Fibre Effects and ISTD Addition

In the present setup, the comparability of different samples, of different experiments, and of the experimental period and the on-line calibration was maintained by the added ISTDs. To validate the temporal stability of the permeation source, the enriched outflow was monitored over a period of one week by PTR-ToF-MS. The RSDs of the PI-normalized responses were as low as 5 % for phenol-$d_6$ and 6 % for acetophenone-$d_8$, confirming the suitability of the setup for on-line ISTD addition. Hence, the variability of the ISTD responses measured by SPME-GC-MS (Fig. 8) is indicative for changes in the preparation

of the derivatization efficiency, the performance of the GC-MS, and most importantly fibre effects.

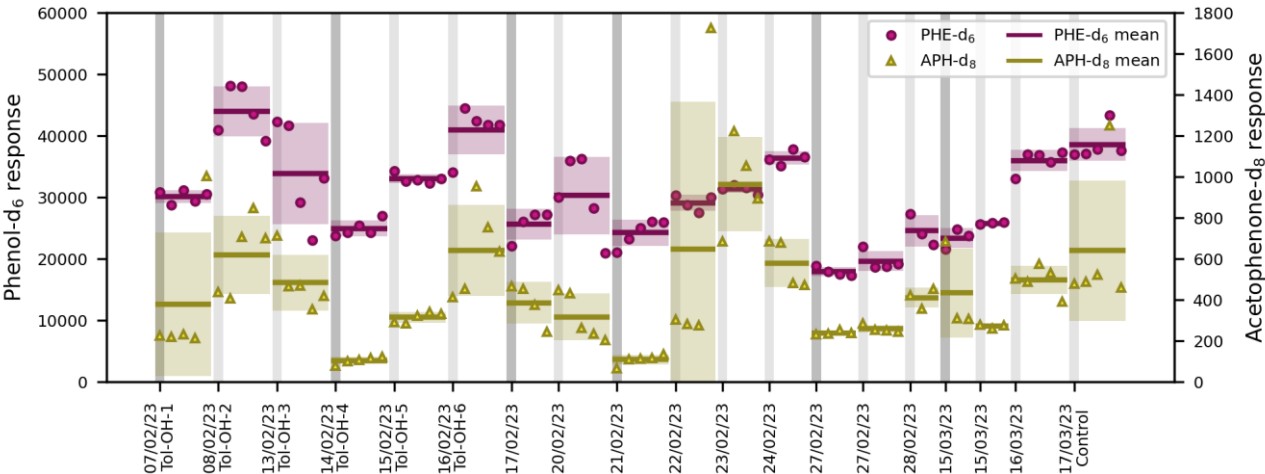

**Figure 8: ISTD responses at the SPME-GC-MS over the entire experimental period in chronological order.** Each individual marker represents one measurement of phenol-$d_6$ or acetophenone-$d_8$, respectively. Only the experimental data are included, not the fibre conditioning and chamber cleanliness test measurements prior to each experiment. The grey vertical bars indicate the start of a new

experiment. The insertion of a new SPME fibre is illustrated in dark grey. The horizontal lines show the mean value of the entire experiment, with the standard deviations included as uncertainty areas.

For SPME sampling, as much as 200 assays per fibre have been reported in the literature (Martos and Pawliszyn, 1997). Here, the number of sampling cycles (PFBHA + on-line sampling + MSTFA/TMCS) conducted with a single fibre was in the range

of about 20 – 30 (Fig. 8) and therefore considerably lower. This high fibre turnover is likely related to (i) the multitude of steps per sampling cycle, (ii) the active analyte extraction where the fibre is located perpendicular to a flow of air with a high velocity of 131 cm s$^{-1}$, and (iii) additional stress caused by the added derivatization reagents, especially TMCS.

The ISTD response often increased over the lifetime of a single SPME fibre (Fig. 8), suggesting either a carry-over effect or

an enhanced performance of the fibres after repeated usage. This development typically reversed after three to four experiments, and fibres were exchanged when this was apparent in the preparatory work of a given experiment. While there was no general trend over time, the SPME fibres differed in their overall extraction efficiency. The mean values of each fibre across all samples in the associated experiments were compared to indicate inter-fibre reproducibility, resulting in relative





standard deviations of 17 % for phenol-$d_6$ and 25 % for acetophenone-$d_8$. This is in accordance with the inter-fibre variability

of 18 % reported previously by Tumbiolo et al. (2004). It is important to note that with the controlled sequencing of the SPME autosampler system, the conditions and the duration of the extraction process were exactly the same for all samples. Thus, reproducibility could be achieved even for pre-equilibrium conditions when time control is critical.

Functional similarity is important for the ability of ISTDs to correct analytes under various conditions, yet just one ISTD was

available for all compounds derivatized by each of the employed derivatization reagents, respectively. Most compounds derivatized by MSTFA/TMCS contained a hydroxy group and were functionally similar to phenol-$d_6$. The concentrations of the two studied carboxylic acids (pyruvic acid, benzoic acid) throughout the photooxidation experiments rarely exceeded their high LODs (Table 3), so that potential errors associated with insufficient resemblance to the assigned ISTD did not become relevant in the data evaluation. All carbonyls derivatized by PFBHA were corrected with acetophenone-$d_8$. As aldehydes are

generally more reactive towards PFBHA than ketones (Jang and Kamens, 1999), acetophenone-$d_8$ is likely to be derivatized less effectively if there is a strong competition for the reagent (Fig. S10). In that case, concentrations of the aldehyde products would be overestimated. We therefore recommend a broader range of more functionally diverse ISTDs for future applications. Nevertheless, even with the available ISTDs and regardless of the class of compounds, we achieved improved fit qualities ($R^2$) of the calibration curves upon normalization of the instrumental response to the assigned ISTD (see calibration curves with

ISTD correction in Fig. 7 and without ISTD correction in Fig. S11). This indicates an advanced method precision.

### 4.2.3 PTR-ToF-MS Calibration and Instrumental Inter-Comparison

The PI-normalized sensitivities, instrumental uncertainties, and LODs of the toluene photooxidation products calibrated for the PTR-ToF-MS are listed in Table S6. Overall, the ring-retaining compounds had the highest instrumental sensitivity. In

contrast, the sensitivity of glyoxal on $m/z$ 59.0041 was very low and barely quantifiable. Firstly, this is due to its low proton affinity of 161.41 – 165.06 kcal mol$^{-1}$ that barely exceeds the proton affinity of water (Wróblewski et al., 2007). Secondly, we observed fragmentation of glyoxal to formaldehyde on $m/z$ 31.0145, described also by Stönner et al. (2017). However, as formaldehyde can occur as a product of toluene photooxidation (Seuwen and Warneck, 1996; Wagner et al., 2003) but could not be determined sufficiently well using the on-line SPME-GC-MS method (Table S9), we had no possibility to differentiate

between glyoxal and formaldehyde on that mass, and therefore measured glyoxal on its more specific protonated mass.

By performing separate calibrations for the cresols and benzyl alcohol, we could determine isomer-specific sensitivities (Table S6). We found the highest sensitivities for $o$-cresol and $m$-cresol, while the sensitivity of $p$-cresol was by about a factor of 2.5 lower. Benzyl alcohol did not show any signal on $m/z$ 109.0626 and thus did not contribute to the sum signal. Instead, it

fragmented to $m/z$ 91.0522, which is in accordance with the literature (Yeoman et al., 2021) and probably indicates the formation of the benzyl radical after abstraction of water from the protonated molecule. The isomer-specific calibration was



performed with relative abundances of 0.71 for *o*-cresol, 0.04 for *m*-cresol, 0.21 for *p*-cresol, and 0.04 for benzyl alcohol (Sect. 4.3.2). As *o*-cresol is by far the most abundant isomer, its contribution to the weighted calibration factor on *m/z* 109.0626 was as high as 87 %. Overall, considering the relative abundances and the specific sensitivities of the individual isomers led to a
1.47-fold increase of the calibration factor compared to the non-weighted calibration (Table S6).

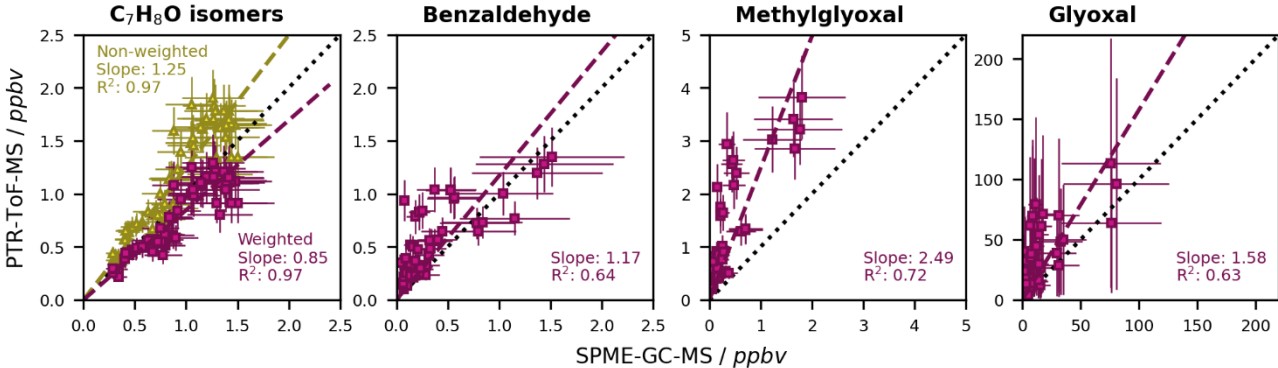

**Figure 9: Comparison of the non-corrected mixing ratios of the toluene first generation products that were calibrated for both the SPME-GC-MS and the PTR-ToF-MS.** The dashed black line represents a 1:1 fit. For both instruments, the mixing ratios as derived from the chamber calibration are shown. In case of the cresol isomers and benzyl alcohol, the non-weighted and weighted calibrations for the sum of the isomers are distinguished. Error bars represent the quantification error of the respective instruments. Error bars of the sum of the
$C_7H_8O$ isomers (cresols and benzyl alcohol) at the SPME-GC-MS are the weighted sum of the individual errors. The data are derived from 18 different experiments at different temperatures and both in the absence and presence of $NO_x$.

The difference between the PTR-ToF-MS data and the SPME-GC-MS data is shown for all calibrated compounds in Fig. 9. For the cresol isomers and benzyl alcohol, the weighted calibration improved the coherence between the two instruments from
25 % to 15 % deviation. Additionally, we determined the best fit quality ($R^2$) for these compounds, which likely relates to the good detectability and relatively low instrumental error of these compounds at both the SPME-GC-MS and the PTR-ToF-MS. The difference between the two instruments was below 20 % for all ring-retaining first generation products, and therefore within the quantification error of both instruments. For the dicarbonyls, on the other hand, there was a substantial instrumental disagreement. While the analysis of glyoxal at both the SPME-GC-MS and the PTR-ToF-MS was too uncertain to reliably
interpret the observed relationship, the concentration of methylglyoxal was clearly higher when derived by PTR-ToF-MS. Possibly, this could be due to the lower selectivity of the PTR-ToF-MS and overestimations due to analytical artefacts. Especially for small molecules, such artefacts can relate to either isobaric interferences, or the actual formation of the target compounds within the instrument upon fragmentation of larger compounds (Michoud et al., 2018; Salazar Gómez et al., 2021). For instance, the $C_4$ and $C_5$ γ-dicarbonyl co-products of glyoxal and methylglyoxal photolyze rapidly in natural sunlight
(Newland et al., 2019) and therefore are likely to produce substantial amounts of follow-up products in our experiments. The MCM suggests the formation of photolytic products with a high degree of oxygenation (Bloss et al., 2005), some of which may well produce $C_3H_5O_2^+$ fragments. Oppositely, underestimated concentrations at the SPME-GC-MS could result for instance from insufficient equilibration in the SPME-GC-MS transfer line (Sect. 4.2.1).





As shown in Fig. 10, the combination of SPME-GC-MS data and weighted PTR-ToF-MS data effectively provides isomeric information on a high temporal resolution. This is particularly valuable for modelling and for gaining insights into chemical mechanisms. To the best of our knowledge, this study marks the first demonstration of a correction of the sum signal based on both the abundance and the sensitivity of the individual isomers. Rather than performing a separate weighted calibration for each time segment assigned to a specific SPME-GC-MS sample, we used a fixed relative abundance of the $C_7H_8O$ isomers for

the weighted calibration and the experimental data. This is a reasonable assumption, as their sinks in the BATCH Teflon chamber (wall losses, photolysis, secondary OH radical reactions) agree within 26 % (Tables S1, S7, S8). The experimental SPME-GC-MS data confirm that the variability of the isomeric distribution over the course of the conducted experiments is only 17 % and hence within the quantification error. In addition to the simplicity of implementation, the adoption of a fixed relative abundance also makes the PTR-ToF-MS results less susceptible to the random error of the SPME-GC-MS data.

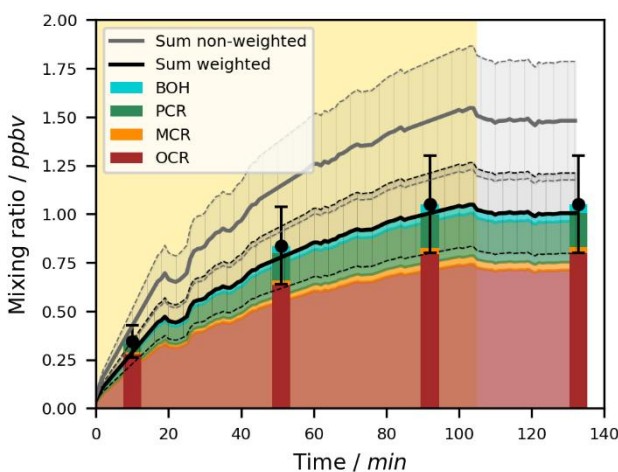


**Figure 10: Time series of the cresol isomers and benzyl alcohol by SPME-GC-MS and PTR-ToF-MS.** Data are shown for experiment Tol-OH-6 (T = 298±1 K, NOₓ-free). No loss corrections were applied to the depicted data. The area shaded in yellow represents the period in which the chamber was irradiated. The sum signal of the PTR-ToF-MS is shown for both the weighted and the non-weighted calibration, with the quantification error shown as shaded area. For the weighted data set, the individual isomers are shown according to their fixed
relative abundances. The error of the sum of the isomers in the SPME-GC-MS data is calculated as the weighted sum of the individual quantification errors. In the depicted experiment, the sum of the isomers agrees exceptionally well between the two instruments, while the measured isomeric distribution deviates slightly from the fixed relative abundance.

**4.3 Application: Product Study**

The conducted experiments allowed us to validate the developed methods and to gain insight into the OH radical-induced photooxidation of toluene. Since the calculated product formation yields take into account compound- and chamber-specific losses and the ambient oxidation efficiency, they provide a bridge from individual experiments to generalizable mechanistic evaluations. Thus, our results can be placed in the context of previous studies and atmospheric chemistry models.





### 4.3.1 Loss Corrections and Error

The contribution of the three characterized loss processes (wall losses, photolysis, secondary OH radical reactions) to the final concentration differed over time and according to the experimental conditions. As shown for the $NO_x$-free experiment Tol-OH-6 in Fig. 11, the different losses gained in importance over the course of the experiment, and the fraction of the measured non-corrected data decreased accordingly. Furthermore, the absolute and relative contributions of the various loss corrections differed for the individual compounds. In the exemplary case of the ring-retaining first generation products, the cresol isomers

and benzyl alcohol were affected mainly by wall losses and secondary reactions with OH radicals, whereas photolytic losses were relatively more important for benzaldehyde.

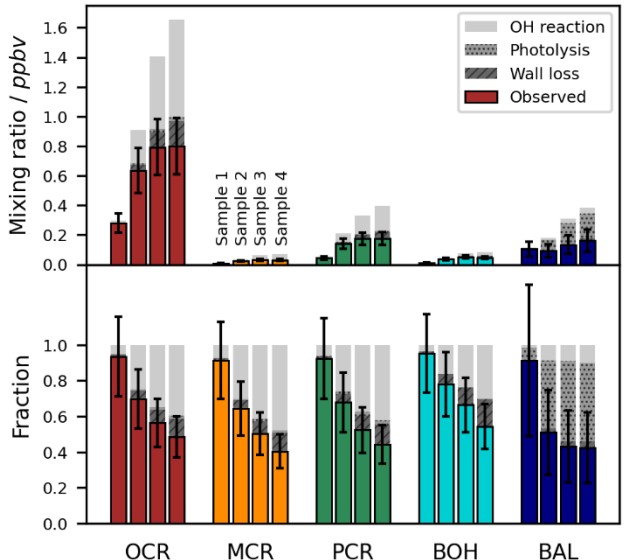

**Figure 11: Contributions of the non-corrected data and each of the three included loss corrections (wall losses, photolysis, secondary OH radical reactions) to the final concentration.** The displayed data refer to the SPME-GC-MS samples 1 to 4 in experiment Tol-OH-6
(T = 298±1 K, $NO_x$-free). The shares of the raw data and the losses are shown on an absolute scale (mixing ratios, upper graph) and on a relative scale (normalized to the total corrected mixing ratio, lower graph). The error bars represent the compound-specific quantification errors of the non-corrected data.

    The loss corrections introduced additional uncertainty in the final data sets. For the wall loss rates, we adopted the uncertainty

of the parameterization (14 %) since we used the modelled values for all compounds. The error of the photolysis rates (38 %) resulted from the recorded spectrum, the uncertainty in the literature values used for the quantum yields and absorption cross sections, and temporal variations in the emission of the solar simulator. The uncertainty associated with secondary reactions of the analytes with OH radicals (19 %) was calculated from the quantification error of toluene, the error of the rate constant of toluene with OH radicals, and the error of the rate constants of OH radicals with the analytes. The total error of the loss-

corrected concentrations was weighted and calculated for each data point separately to account for the variable contribution of the different loss processes (for details, see Supplement S7 with Table S10).



As proof-of-concept for the implemented correction scheme, we furthermore compared the measured and the loss-corrected data sets obtained by SPME-GC-MS and PTR-ToF-MS for different compounds and experiments (see Supplement S7 with Fig. S12 and S13). The absolute losses for each of the distinct loss processes as well as the total loss-corrected concentrations are in good agreement between the SPME-GC-MS and the PTR-ToF-MS data, indicating that the correction procedure is valid for both instruments in spite of their different temporal resolutions.

### 4.3.2 Formation Yields

The linear regressions performed for deriving the formation yields of the ring-retaining first generation products of toluene for $T = 298\pm1$ K in the absence of $NO_x$ are shown in Fig. 12. For demonstration, and to be able to resolve the $C_7H_8O$ isomers, we focus here mainly on the SPME-GC-MS data. The results obtained in this study are clustered according to the reaction channel and compared with the MCM predictions and literature values in Table 4.

For the cresol channel, we derived a sum yield of 10.8±2.5 %. This is in the low range of most reported values, but in close agreement with the 10 % yield obtained in another recent study by Zaytsev et al. (2019). The large discrepancy of the available literature values (9 − 52.9 %) can be explained by the high reactivity of the cresols and by their susceptibility to various physical and chemical losses that in turn are specific to the experimental setup and analytical technique (Klotz et al., 1998; Schwantes et al., 2017). Notably, the three highest cresol yields reported in the literature in the last 20 years were obtained theoretically (Wu et al., 2014) or under low-$O_2$ conditions (Baltaretu et al., 2009; Ji et al., 2017) that may disfavour the competing dicarbonyl channel (Jenkin et al., 2009; Newland et al., 2017). The very high yield of 52.9 % obtained by Seuwen and Warneck (1996) was obtained from experiments with toluene mixing ratios of about 1000 ppmv, limiting the transferability of these data to the real atmosphere. The isomeric distribution was previously reported in the order $o$-cresol > $p$-cresol > $m$-cresol (Klotz et al., 1998; Seuwen and Warneck, 1996; Smith et al., 1998). This pattern is well reproduced in the present work with yields of 8.0±1.8 %, 2.4±0.6 %, and 0.4±0.1 %, respectively. These yields correspond to a ratio of 74:4:22 for $o$-, $m$-, and $p$-cresol. As Seuwen and Warneck (1996) could not quantify the yield of $m$-cresol, and other studies evaluated the sum of $m$- and $p$-cresol due to insufficient chromatographic separation (e.g. Moschonas et al., 1998), this study reports the first $NO_x$-free distribution of all the cresol isomers to the best of our knowledge.

The yield of benzyl alcohol determined in this work (0.5±0.1 %) is within the range of the few reported values (Moschonas et al., 1998; Seuwen and Warneck, 1996; Smith et al., 1998), confirming it as a relevant first generation product in the abstraction channel. While the yield of benzyl alcohol was always found to be lower than that of benzaldehyde, it is uncertain whether it is more abundant (Seuwen and Warneck, 1996), less abundant (Smith et al., 1998), or in a similar range (this work) compared to its isomer $m$-cresol. This is important for the interpretation of purely mass-spectrometric approaches that often neglect benzyl alcohol and assign the sum signal exclusively to the $o$-, $m$-, and $p$-isomers of cresol (e.g. Baltaretu et al., 2009; Ji et al.,




830  2017; Zaytsev et al., 2019). While we found that benzyl alcohol does not contribute to the sum signal in $H_3O^+$ soft ionization, its sensitivity in other ionization techniques should be verified to avoid overestimated cresol yields.

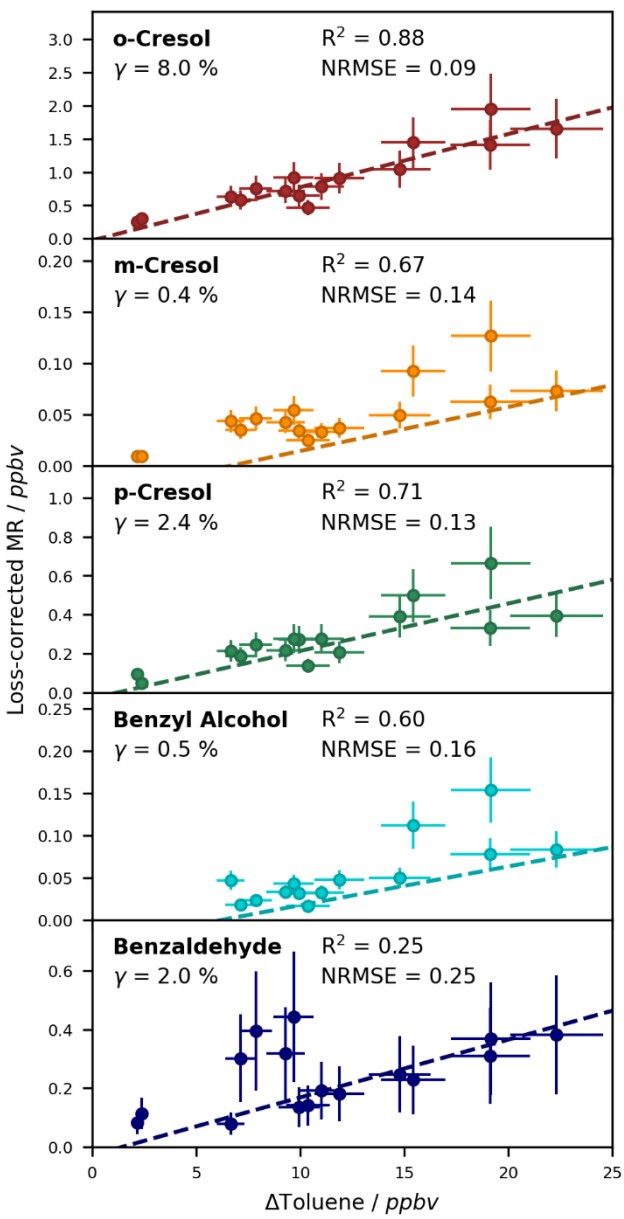

**Figure 12: Yields of the ring-retaining first generation products of toluene, derived by linear regressions of the loss-corrected mixing**
835  **ratio (including loss corrections for wall losses, photolysis, secondary OH radical reactions) against the monitored change in toluene.**
For each compound, the slope of the regression is equal to the formation yield. To perform the regressions, only experiments conducted at T = 298±1 K and in the absence of $NO_x$ were used. Error bars represent the quantification error.



**Table 4: Yields of the ring-retaining first generation products of toluene.** The compounds are clustered according to the main channel, which in turn is listed as the sum of the associated products. All yields are given in %. Unless stated otherwise, the yields shown for this work are based on the SPME-GC-MS data. The error of these yields was calculated from the propagation of the NRMSEs of the linear regressions, the calibration error, and the quantification error of toluene. The yields in the MCM refer to the latest representation of the toluene chemistry in MCM v3.1 (Bloss et al., 2005). The MCM does not distinguish between the cresol isomers, while in the abstraction channel, the product yields depend on the chemistry of the preceding peroxy radical. Underlined references refer to $NO_x$-free studies, while all other cited yields were obtained in the presence of NOx. The asterisk * indicates computational values, as presented by Wu et al. (2014). The instrumental techniques used in the cited literature were chemical ionization mass spectrometry (Baltaretu et al., 2009; Ji et al., 2017; Zaytsev et al., 2019), differential optical absorption spectroscopy (Klotz et al., 1998), gas chromatography with flame ionization detector (Atkinson et al., 1983, 1989; Atkinson and Aschmann, 1994; Moschonas et al., 1998; Seuwen and Warneck, 1996; Smith et al., 1998), offline determination of carbonyls after derivatization (Seuwen and Warneck, 1996; Smith et al., 1998), and tandem mass spectrometry (Dumdei et al., 1988). The literature values presented here provide a comprehensive overview of relevant data; however, we acknowledge that there may be additional relevant studies beyond those included in this table.

| Channel, Compound | Formation yields in % | | | |
|---|---|---|---|---|
| | This work | MCM | Literature | References |
| Cresol | 10.8±2.5 | 18 | 9−52.9 | 9 (Moschonas et al., 1998), 10 (Zaytsev et al., 2019), 17.9 (Smith et al., 1998), 17.9 (Klotz et al., 1998), 25.2 (Atkinson et al., 1989), 28.1 (Baltaretu et al., 2009), 32.0* (Wu et al., 2014), 39.0 (Ji et al., 2017), 52.9 (Seuwen and Warneck, 1996) |
| OCR | 8.0±1.8 | n.a. | 6−38.5 | 6 (Moschonas et al., 1998), 12.0 (Klotz et al., 1998), 12.3 (Smith et al., 1998), 12.3 (Atkinson and Aschmann, 1994), 13.1 (Atkinson et al., 1983), 20.4 (Atkinson et al., 1989), 38.5 (Seuwen and Warneck, 1996) |
| MCR | 0.4±0.1 | n.a. | ≤0.4 −2.7 | ≤0.4[(a)] (Seuwen and Warneck, 1996), 2.6 (Smith et al., 1998), 2.7 (Klotz et al., 1998) |
| PCR | 2.4±0.6 | n.a. | 3.0−14.4 | 3.0 (Smith et al., 1998), 3.2 (Klotz et al., 1998), 14.4 (Seuwen and Warneck, 1996) |
| Abstraction[(b)] | ≥2.5±0.7 ≥5.1±1.8[(c)] | 7 | ≥6.15−9.4 | ≥6.15 (Smith et al., 1998), ≥7.7 (Seuwen and Warneck, 1996), ≥9.4 (Moschonas et al., 1998) |
| BOH | 0.5±0.1 | n.a. | 0.15−2.4 | 0.15 (Smith et al., 1998), 1 (Moschonas et al., 1998), 2.4 (Seuwen and Warneck, 1996) |
| BAL | 2.0±0.6 4.6±1.7[(c)] | n.a. | 4.9−11.3 | 4.9 (Baltaretu et al., 2009), 5.0 (Dumdei et al., 1988), 5.3 (Seuwen and Warneck, 1996), 5.8 (Klotz et al., 1998), 6.0 (Smith et al., 1998), 6.45 (Atkinson et al., 1989), 7.0* (Wu et al., 2014), 7.3 (Atkinson et al., 1983), 8.4 (Moschonas et al., 1998), 9 (Zaytsev et al., 2019), 11.3 (Ji et al., 2017) |

[(a)] *Without correction for secondary losses.*
[(b)] *The MCM v3.1 yield encompasses all possible products of the abstraction channel. For the values obtained in this work as well as for the cited literature values, the sum of only benzyl alcohol and benzaldehyde is compiled.*
[(c)] *PTR-ToF-MS data set for benzaldehyde.*

As the dominant product in the H abstraction channel, benzaldehyde has been studied for many decades. The yield obtained in this work using the SPME-GC-MS data (2.0±0.6 %) is smaller than any of the reported values both in the presence and in the absence of $NO_x$. Notably, the regression performed for benzaldehyde has a relatively high uncertainty (NRMSE = 0.25,





Fig. 12), which is in line with the discussed instrumental variability of the aldehydes determined with the on-line SPME-GC-MS method in our setup. As there are no relevant isomers or spectral interferences for benzaldehyde, we could refer to the PTR-ToF-MS measurements as well. These data resulted in a yield of 4.6±1.7 % (Fig. 13). The fact that this value is by more than a factor of 2 higher than the yield obtained using the SPME-GC-MS data relates to the observed difference in the measured concentrations (Fig. 9) that is further amplified by the applied loss corrections. The PTR-ToF-MS yield for benzaldehyde is in

better agreement with the literature values, albeit still at the lower end. We conclude that in case of benzaldehyde, the SPME-GC-MS data set is too small to compensate for the instrumental error, and that the more precise PTR-ToF-MS data should therefore be used. Interestingly, the PTR-ToF-MS yield agrees reasonably well with the apparent outliers in the SPME-GC-MS data set in the range of $\Delta$toluene = 7.0 – 9.8 ppbv (Fig. 12). These data correspond to the experiments Tol-OH-4 and Tol-OH-5, which in turn show the best intra-experimental stability of acetophenone-$d_8$ (Fig. 8). Although we lack definite proof

that these effects are causally related, this observation nevertheless reinforces our recommendation to include internal standards for on-line analytical methods used in chamber experiments, and to select compounds with good stability and high resemblance to the analytes for correction.

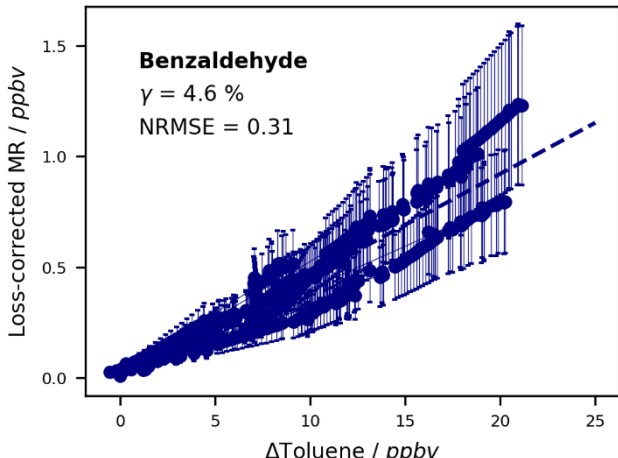

**Figure 13: Benzaldehyde formation yield, derived by linear regression of the loss-corrected mixing ratio (including loss corrections for wall loss, photolysis, and reaction with OH radicals) against the monitored change in toluene.** The slope of the regression is equal to the formation yield. To perform the regression, only experiments conducted at T = 298±1 K and in the absence of NO$_x$ were used. Error bars represent the uncertainty of the loss-corrected data.




**5 Conclusion**

In this work, we set up an indoor atmospheric simulation chamber to study multifunctional photooxidation products in the gas phase by on-line SPME-GC-MS and PTR-ToF-MS. Specifically, the products emerging from the reaction of toluene with OH radicals were investigated and used as reference compounds to characterize the chamber properties and the instrumental
capabilities. In the new chamber, controlled experiments could be performed with realistic photon fluxes and OH radical concentrations. We characterized the photolysis rates ($1.77{\times}10^{-8} - 3.02{\times}10^{-4}$ s$^{-1}$), wall losses ($4.54{\times}10^{-6} - 8.53{\times}10^{-5}$ s$^{-1}$), and secondary OH radical reactions of photooxidation products with diverse functional groups. The wall losses were determined empirically, and found to be dependent on fundamental molecular properties. We parameterized the loss rates accordingly, opening up the possibility of accurately characterising the losses of compounds for which no authentic standards are available.
In addition to the wall losses, we calculated photolytic and reactive losses using available literature values of the absorption cross sections and quantum yields, and the rate constants for the reactions with OH radicals, respectively.

The SPME-GC-MS method with on-fibre PFBHA and MSTFA derivatization enabled the on-line analysis of fragile oxygenated compounds with high sensitivity and isomeric resolution. Due to the continuous addition of functionally similar
internal standards from a customized permeation source, variations in the SPME fibre extraction efficiency were effectively accounted for. Our method worked particularly well for the studied aromatic alcohols. The carbonyl compounds were associated with a higher instrumental error due to the limitations of the experimental setup regarding the on-line analysis of sticky compounds. For future applications, we recommend a wider range of internal standards to facilitate a higher structural resemblance to the analytes. Combining the SPME-GC-MS data with the PTR-ToF-MS data, we could gain additional insight
into temporal trends and cross-compare the absolute concentrations. We calibrated the main first generation products of toluene for both instruments, and found good agreement for the ring-retaining products (the cresol isomers, benzyl alcohol, benzaldehyde). We further demonstrated that the sensitivity of the PTR-ToF-MS for different structural isomers can vary substantially. For the cresols and benzyl alcohol, we therefore applied a weighted calibration factor based on the determined isomer-specific sensitivities and the distribution of the isomers as obtained by the SPME-GC-MS data. This led to an improved
instrumental agreement, and allowed us to derive continuous time series of the individual isomers. For a wide range of applications based on mass-spectrometric techniques, it may be worth examining and potentially incorporating isomer-specific sensitivities if the relative distribution is known and calibration standards are available.

For method validation purposes, we focused on the initial photooxidation period to evaluate the primary chemistry and the
dominant first generation products of toluene. Taking into account compound-specific physical and chemical losses, we obtained product formation yields under NO$_x$-free conditions at T = 298±1 K for *o*-cresol (8.0±1.8 %), *m*-cresol (0.4±0.1 %), *p*-cresol (2.4±0.6 %), benzyl alcohol (0.5±0.1 %), and benzaldehyde (4.6±1.7 %). These results are comparable to previous studies. Jointly, the experimental design, instrumental analysis, and data processing approaches that we have tested and



optimized in this work provide a means of determining the formation yields of the individual isomers of photooxidation

products with different functional groups with good accuracy.

**Code and data availability**

The code and data used in this study are available by request to the corresponding authors.

**Author contributions**

AN acquired funding, conceptualised the project, and provided supervision. EB and AM provided methodological training at the EUPHORE chamber. With support from AN, FL set up the methodology in Bayreuth, performed the experiments, validated the results, and was in charge of the curation, formal analysis, and visualisation of the data. FL and AN prepared the manuscript with contributions from all co-authors.

**Competing interests**

The authors declare that they have no conflict of interest.

**Acknowledgements**

We thank the workshop of the University of Bayreuth for manufacturing the SPME-GC-MS sampling cell. Also, we thank Andrej Einhorn and Agnes Bednorz for their support regarding the technical setup and the experimental work. We thank Sergej

Bleicher for helpful discussions about the solar simulator emission spectrum, and Cornelius Zetzsch for advice regarding hydroxyl radical generation. We are grateful for the financial support of the German Research Foundation (Deutsche Forschungsgemeinschaft, DFG) for the PTR-ToF-MS (Großgeräteantrag, project number: 495692966) and the ACTRIS TNA for practical training at the EUPHORE chambers (supported by the European Commission under the Horizon 2020 – Research and Innovation Framework Programme, H2020-INFRADEV-2019-2, Grant Agreement number: 871115). This open access

publication was funded by the Open Access Publishing Fund of the University of Bayreuth.





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
