# Peer review of "Characterization of the new BATCH Teflon chamber and on-line analysis of isomeric multifunctional photooxidation products"

_EGUsphere, 2024_

## Referee Comment (RC1)

Löher et al. describe the characterization of a new atmospheric simulation chamber built at the University of Bayreuth. The authors describe the design, photolysis, chemistry, and wall interactions of the chamber for purposes of gas-phase mechanism studies. The authors also describe SPME and PTR-ToF-MS techniques deployed to study important intermediates of VOC oxidation. The authors study the OH oxidation of toluene as a testbed for the new chamber and to demonstrate the capabilities of the system in quantifying product yields.

The paper is very well-written and organized and provides all of the necessary information to understand the performance of the BATCH reactor. I view this manuscript as a reference point for future studies utilizing the BATCH reactor, as is typical for other chamber and/or flow tube description papers published in AMT or similar journals. The manuscript also provides some nice science describing the chemistry of toluene and yields for relevant products. I only have one substantive comment and a small number of minor comments and suggestions. I support publication.

**Main comment**

**Section 4.1.2.** The authors parameterize measured wall losses using simple equations. How does this parameterization compare against theoretical considerations or other efforts to quantify wall losses using chemical information? There are a number of papers that have investigated vapor wall loss and developed a number of parameterizations (e.g., Yeh and Ziemann, Ye et al. 2016, McVay et al. 2014 to name a few). These are estimated using C*, which in itself is related to molecular weight by the ideal gas law. How do these relate to the functions described here? And why the square root of the vapor pressure? Some context and description of previous parameterizations would be helpful.

I can see how this is a valid approach for the toluene system, but is it the intention of the authors to use this more broadly for other systems? If so, I would think that it may be necessary to discuss this in more detail and how this parameterization compares to other forms (as noted in comment above). This parameterization seems very simple compared to previous efforts to characterize wall loss rates, but I may not be aware of all of the various efforts.

https://www.tandfonline.com/doi/full/10.1080/02786826.2016.1195905

https://www.tandfonline.com/doi/full/10.1080/02786826.2015.1068427

https://pubs.acs.org/doi/10.1021/es502170j

**Minor Comments**

**Lines 101-112:** This is a very nice discussion about the utility of derivatization.  Can the authors point to some relevant studies on the aromatic systems that have used

derivatization to quantify the yields of aldehydes, acids, etc.? I think this would be helpful to place in context why derivatization is a powerful technique and relevant to the studied toluene system.

**Lines 125-132:** I would also point towards papers that have studied the SOA and product yields from the second generation products (e.g., cresol, Schwantes et al. 2017). This would be relevant to the discussion that follows about multi-generational formation of glyoxal.

**Line 154:** This is very impressive temperature range I may have missed this in the text, but what are the dimensions of the containment room? How quickly does it take to reach the desired temperature set piont? Is it possible to do dynamic temperature changes during experiments? My take is that this system is about the same size as that described by Osseiran et al. and contained in a similar temperature-controlled vessel. Is that correct? As a note, it might be nice to have a picture of the chamber in a TOC graphic to get a sense of the scale.

https://www.sciencedirect.com/science/article/abs/pii/S1001074220301170

**Line 168**: Is this the footprint of the Teflon chamber? Can the third dimension be provided to get a sense of the full size at max volume (300 L)?

**Line 250**: Perhaps useful to note the estimated mass that could be collected on the SPME fibers for a typical experiment (e.g., the toluene system)?

**Line 256-262**: Later in the text, the authors note that the stability of ISTDs were tracked for a week using PTR-ToF-MS. I would include that discussion here.

**Lines 382-385:** Was there an order that experiments were conducted? I.e., were NOx-free experiments conducted prior to those where NO was injected? If not, is there uncertainty associated with possible NOx sources from the walls? Previous studies have shown that walls act as a NOx source once NO or NO2 are introduced as reactants (e.g., Rohrer et al.), so I feel that this should be referenced here.

https://acp.copernicus.org/articles/5/2189/2005/acp-5-2189-2005.html

**Line 419:** Do wall losses include possible line losses?

**Line 502:** For consistency with the previous comparison, I would recommend phrasing that the experimental photolysis rate is "21% higher" than the calculated value.

**Line 525-527:** This sentence tends to suggest that HONO is not contributing, but this is a little speculative without HONO or measurements of NO / NO2 at low detection limits. I would suggest leaning on the conclusion that H2O2 is the main source of OH radicals and leave out the mention of HONO.

**Lines 529-531:** Here, and elsewhere, it would also be helpful to see the photolysis and OH lifetimes for each species. This would help the reader to gauge how much loss of these species might be expected after they are formed relative to OH or other processes and complement the loss distributions shown in Fig. 11.

**Figure 8:** It is somewhat difficult to see the dark grey bars relative to the lighter grey bars. Perhaps using a different color would help? Or, dotted vertical lines could indicate new experiments?

Lines 752-753: This may be the case for chamber studies, but previous studies investigating field observations of isomer distributions (e.g., C8-aromatics, propanal / acetone) have applied isomer-specific sensitivities to interpret PTR signals in very complex mixtures (e.g., biomass burning, Koss et al.). This is also a technique currently used in other mass spectrometers with significant variability in isomer sensitivities (e.g., I-CIMS, Xiong et al. )

https://acp.copernicus.org/articles/15/11257/2015/

https://acp.copernicus.org/articles/15/11257/2015/18/3299/2018/

---

## Author Comment (AC1)

We are grateful for the reviewer's critical review and suggestions. In the following, we addressed the specific comments one by one. The corresponding changes in the text are highlighted in a marked-up version of the manuscript accordingly. Hereinafter, we structure our responses to the comments in the following sequence:

**Referee's comments/questions in bold font.**
Author's response in regular font.
*Proposed changes in italic font.*

Main comments

**General**
**I think, this paper and your (and your colleagues') future work would benefit from stating, which of the characterizations need to be performed regularly, before every experiment and which data can be re-used from your publication as-is. I suggest to describe a "best-practice procedure" in this paper, that you can refer to in the future.**

We agree that such a paragraph would be useful and added it to the manuscript as section 4.1.4 (within the chamber characterization chapter in the results/discussion).

Added text:

*4.1.4 Best Practice for Future Work*

*For future work, we propose a best practice procedure to re-evaluate chamber properties when experimental conditions change and time progresses.*

- *The qualitative pattern of the solar simulator emission spectrum is stable over time, yet we found a difference in the absolute photon flux by about a factor of 3 over the course of roughly one decade (Fig. 4a). Therefore, the spectrum should be recorded regularly. More frequent checks are needed when there is high experimental throughput or when new HMI lamps are inserted, since each lamp is unique and metal depositions on the glass lead to a relatively quick loss of intensity at the beginning of the lamp lifetime (Bleicher, 2012).*
- *Wall losses need to be re-assessed (i) for experiments operated at different temperatures, (ii) for experiments performed in humid air, (iii) for new VOC photooxidation systems with different compound structures and properties, (iv) for new chambers when the Teflon body was replaced (see below), and (v) for longer periods of time and more complex wall interactions if the experimental duration is increased.*
- *Prior to every single experiment, the chamber needs to be carefully cleaned and evaluated for possible artefacts and contamination by recording chamber blank values. For each set of experiments, at least one control experiment without the VOC precursor should be performed. As the chamber is completely evacuated at the end of any experiment, it should still be empty when the next experiment is started. If this is not the case, leaks in the chamber body need to be fixed. A new Teflon body should be constructed after working with high concentrations of sticky compounds and/or $NO_x$ to avoid carry-over or the unintended formation of photochemically relevant species such as HONO (Bell et al., 2023; Rohrer et al., 2005).*

The following paper was added to the bibliography:

*Bell, D., Doussin, J.-F., and Hohaus, T.: Preparation of Simulation Chambers for Experiments, in: A Practical Guide to Atmospheric Simulation Chambers, edited by: Doussin, J.-F., Fuchs, H., Kiendler-Scharr, A., Seakins, P., and Wenger, J., Springer International Publishing, Cham, 113–127, https://doi.org/10.1007/978-3-031-22277-1_3, 2023.*

**UV part of the bandpass-filtered solar simulator spectrum (section 4.1.1)**
**As you measure ozone, you could use the photolysis of ozone alone in the chamber to double-check the scaling of the old UV-spectrum. Currently, the scaling of the old dataset is motivated by the comparison with the new dataset at higher wavelength, but it is not clearly tested, if the simple scaling of the old data is correct.**

With this comment, you highlight a technical issue that we came across because the new spectroradiometer does not completely overlap with the range of wavelength of the DOAS, which was used previously, leaving a gap in the lower UV-range ($\lambda$ = 262 – 325 nm). Instead of only reporting the new spectrum, we decided to make use of the previously recorded spectrum, as the qualitative features were very similar in both the old and new spectrum. To check the validity of the proposed scaling of the old dataset for the lower UV-range, we performed two actinometric experiments with $NO_2$ and methylglyoxal, respectively.

Here, you propose to use the photolysis of ozone as an additional check. This requires quenching of the formed $O(^1D)$ to suppress the formation of $O(^3P)$ and the ultimate re-formation of $O_3$ via the reaction of $O(^3P)$ with $O_2$ (e.g. Müller et al., 1995, https://agupubs.onlinelibrary.wiley.com/doi/epdf/10.1029/95GL00203). Due to the risk for carry-over and contamination, we prefer not to add large quantities of $O(^1D)$ reagents such as $N_2O$ into our chamber. However, in our $N_2/O_2$ matrix, the $O_3$ photolysis rate is very difficult to evaluate due to the cycling of $O_3$, $O_2$, $O(^1D)$, and $O(^3P)$. With all these reactions, the photolysis rate of ozone would need to be calculated by a chemical box model, and would be associated with a high uncertainty due to the numerous assumptions that would play into the final result, especially so as the main $O_3$ photolysis channel switches from direct $O(^1D)$ production to direct $O(^3P)$ production within the spectral range of the scaled spectrum.

We therefore believe that our $NO_2$ and methylglyoxal actinometric experiments serve the purpose of validating the (scaled) spectrum sufficiently. We realize that our text might have not been clear enough on the meaning of these actinometric experiments. We therefore made the following changes:

Firstly, we adapted the method part.

Old version (section 3.1.1, last paragraph):
*To validate the recorded spectrum, we performed two actinometric experiments.*

New version (section 3.1.1, last paragraph):
*To validate the recorded spectrum as well as the scaled spectrum in the UV range, we performed two actinometric experiments.*

In the Results/Discussion section, we already included a comparison of the calculated and empirical $NO_2$ and methylglyoxal photolysis rate. We now highlighted the qualitative resemblance of the old and the new spectrum, and also added a statement to contextualise this comparison and to make an assessment of the validity of the spectrum (including the scaled spectrum).

Old version (Sect. 4.1.1, 1st paragraph):
*This difference could relate to the specific emission and age of each lamp, or to the vertical distance at which the previous spectrum was recorded. Using the newly measured spectrum ($\lambda$ = 325 – 1000 nm) and the spectrum scaled from the old data set ($\lambda$ = 262 – 325 nm), the $NO_2$ photolysis rates calculated theoretically ($2.07\pm0.79\times10^{-2}$ $s^{-1}$) and derived experimentally ($2.32\pm1.99\times10^{-2}$ $s^{-1}$, Fig. S2) agree within 12 %. For methylglyoxal, the experimental photolysis rate ($3.66\pm0.56\times10^{-4}$ $s^{-1}$, Fig. S3) is 21 % higher than the calculated value ($3.02\pm1.15\times10^{-4}$ $s^{-1}$), yet the uncertainties of these rates overlap.*

New version (Sect. 4.1.1, 1st paragraph):
*This difference could relate to the specific emission and age of each lamp, or to the vertical distance at which the previous spectrum was recorded. Meanwhile, the same qualitative features are visible in both spectra. Using the newly measured spectrum ($\lambda$ = 325 – 1000 nm) and the spectrum scaled from the old data set ($\lambda$ = 262 – 325 nm), the $NO_2$ photolysis rates calculated theoretically ($2.07\pm0.79\times10^{-2}$ $s^{-1}$) and derived experimentally ($2.32\pm1.99\times10^{-2}$ $s^{-1}$, Fig. S2) agree within 12 %. For methylglyoxal, the experimental photolysis rate ($3.66\pm0.56\times10^{-4}$ $s^{-1}$, Fig. S3) is 21 % higher than the calculated value ($3.02\pm1.15\times10^{-4}$ $s^{-1}$), yet the uncertainties of these rates overlap. Hence, the updated spectrum with the scaled data in the UV range allows us to reliably derive photolysis rates in the BATCH Teflon chamber.*

With this change, we feel that the structure of the following paragraphs within section 4.1.1 would be more intuitive if the photolysis rates of the analytes are mentioned next. Therefore, we changed the order of the paragraphs as follows:

Old version (sect. 4.1.1 structure):
1) *"The measured emission spectrum ..."*
2) *"By design, ..."*
3) *"The emission of the solar simulator ..."*
4) *"The calculated photolysis rates ..."*

New version (sect. 4.1.1 structure):
1) *"The measured emission spectrum ..."*
2) *"The calculated photolysis rates ..."*
3) *"The emission of the solar simulator ..."*
4) *"By design, ..."*

**Wall losses (section 4.1.2):**
**Since your chamber can be temperature-controlled over a large temperature-range, I suggest to test wall losses at different temperatures (shift of vapor pressure – this would be nice to set into relation with existing parametrizations), but also at different humidities (effect of hygroscopicity can play a role here). Also, since your measured wall losses seem to not level off, I would also suggest to test the maximal wall loss rate with sulfuric acid.**

We agree that temperature, humidity, and time/equilibration all play an important role for wall loss rates in chamber experiments in general terms. For this study, however, these effects have limited relevance: (i) to determine formation yields from loss-corrected concentrations, we only evaluated the experiments at 298 K, which is also the temperature for which we have determined the wall loss rates, (ii) all experiments were performed in dry air, (iii) all experiments were short enough to describe losses to the wall as a continuous process.

We realize that the temperature, humidity, and experimental duration might change in future applications. However, we feel that it is beyond the scope of this paper to address the effect of all these parameters on the wall loss rates, and that this would be a paper on its own. Still, we acknowledge that wall losses need to be re-defined for other temperatures or humidity levels if such experiments are to be performed/evaluated in the future. We added a statement addressing this need in the "best practice" paragraph which we included in response to the first comment. Also, we specifically pointed out that the wall losses that we describe in this work are valid only for the conditions as described above (dry, T = 298 K, short duration).

Old version methods (3.1.2):
*To obtain the individual wall losses, a solution containing all authentic standards was injected into the chamber in a $N_2$ matrix (c = 12.3×10$^{10}$ molecules cm$^{-3}$, see SPME-GC-MS calibration in Sect. 3.2.1).*

New version methods (3.1.2):
*To obtain the individual wall losses, a solution containing all authentic standards was injected into the chamber in a dry $N_2$ matrix at T = 298±1 K (c = 12.3×10$^{10}$ molecules cm$^{-3}$, see SPME-GC-MS calibration in Sect. 3.2.1).*

Old version results/discussion (4.1.2):
*The wall loss rates which were determined empirically in this work (Table S8) ranged between 4.54±1.80×10$^{-6}$ s$^{-1}$ (benzaldehyde) and 8.53±0.68×10$^{-5}$ s$^{-1}$ (p-hydroxybenzaldehyde).*

New version results/discussion (4.1.2):
*The wall loss rates which were determined empirically in this work for dry air at T = 298±1 K (Table S8) ranged between 4.54±1.80×10$^{-6}$ s$^{-1}$ (benzaldehyde) and 8.53±0.68×10$^{-5}$ s$^{-1}$ (p-hydroxybenzaldehyde).*

Added text results/discussion (end of section 4.1.2):
*It should be noted that the presented wall loss rates as well as their parameterization are valid only for dry conditions and a temperature of 298 K, but may differ if experimental conditions change (Grosjean, 1985; Zhang et al., 2015). Furthermore, both the time frame over which we determined the wall losses empirically and the conducted photooxidation experiments were limited to three to four hours. On longer time scales, desorption processes, wall saturation, and equilibrium between the gas phase and the chamber walls need to be taken into account more specifically (Yeh and Ziemann, 2015; Zhang et al., 2015).*

Added text within new best practice part (section 4.1.4):
*Wall losses need to be re-assessed (i) for experiments operated at different temperatures, (ii) for experiments performed in humid air, (iii) for new VOC photooxidation systems with different compound structures and properties, (iv) for new chambers when the Teflon body was replaced (see below), and (v) for longer periods of time and more complex wall interactions if the experimental duration is increased.*

Minor comments

**ll. 120- … : Please give a temperature range for which given toluene + OH reaction rate and abstraction/addition channel ratio are valid**

We added the temperature for the initial reaction of toluene with OH radicals.

Old version (Sect. 2, 1$^{st}$ paragraph):
*Toluene reacts with OH radicals at a rate of about k = 5.6×10$^{-12}$ molecules$^{-1}$ cm$^3$ s$^{-1}$ (IUPAC, 2024) either via addition of the OH radical to the aromatic ring structure or via H abstraction from the substituted methyl group.*

New version (Sect. 2, 1$^{st}$ paragraph):
*Toluene reacts with OH radicals at a rate of about k = 5.6×10$^{-12}$ molecules$^{-1}$ cm$^3$ s$^{-1}$ at T = 298 K (IUPAC, 2024), either via addition of the OH radical to the aromatic ring structure or via H abstraction from the substituted methyl group.*

We agree that the different Arrhenius behaviours of the addition and abstraction channel will result in a dependence of the channel ratios on temperature, and that this should not be simplified without further explanation. As we do not focus on temperature variations in this paper and the text is already quite long, we prefer to not go into too much detail here, and would instead simply delete this sentence, and focus on the MCM representation which likewise implies a dominance of the addition channel.

Old version (Sect. 2, 1$^{st}$ paragraph):
*The addition pathway is dominant with a branching ratio in the range of 0.85 – 0.93 (Atkinson et al., 1980; Hu et al., 2007; Wu et al., 2014). The MCMv3.1 (Bloss et al., 2005) distinguishes between four channels in the primary chemistry of toluene (Fig. 1).*

New version (Sect. 2, 1$^{st}$ paragraph):
*The MCMv3.1 (Bloss et al., 2005) distinguishes between four channels in the primary chemistry of toluene (Fig. 1).*

**Table S1: table S1 also needs the name of the species (at least in the table description), not just the short form, as these short forms were not introduced yet. Not in the main text until this point, nor in the SI.**

We agree that this information is missing in the SI up to this point. As the space in the table is already limited, we instead added a note explaining the full name of the species and their abbreviations at the very beginning of the SI.

Added text (SI, between TOC and Section S1):
*Hereinafter, the following abbreviations are used for toluene photooxidation products and their internal standards (here sorted according to their retention time, as in Table 1 in the main text): phenol-$d_6$ (PHE-$d_6$), o-cresol (OCR), m-cresol (MCR), p-cresol (PCR), benzyl alcohol (BOH), o-nitrotoluene (ONT), (nitromethyl)benzene (NMB), m-nitrotoluene (MNT), benzoic acid (BAC), p-methylcatechol (PMC), glycolaldehyde (GAL), nitrocresols (NCR), pyruvic acid (PAC), acetophenone-$d_8$ (APH-$d_8$), benzaldehyde (BAL), glyoxal (GLY), methylglyoxal (MGL), p-hydroxybenzaldehyde (PHB).*

**l. 230: here I wondered, how long is one experiment roughly? Information on this comes much later in the manuscript, but would be interesting already at a much earlier stage, e.g. here.**

We agree that the information about the total experimental duration (incl. both irradiated and dark periods) comes too late in the manuscript and is only indirectly mentioned (such as in Fig. 3). We therefore included a statement about the total experimental duration in the caption of Table 2 in the method section where the experiments are first described.

Old version (caption Table 2):
*The timing of the first (and all consecutive) SPME-GC-MS samples was varied to better constrain formation yields.*

New version (caption Table 2):
*The timing of the first (and all consecutive) SPME-GC-MS samples was varied to better constrain formation yields. For all experiments, we obtained 4 SPME-GC-MS samples, resulting in total experimental durations of about three hours.*

**l. 265: "SIM" has not been explained yet**

The abbreviation "SIM" has been explained a bit earlier on the previous page ("The GC-MS was operated with splitless injections, a standard HP-5MS column ramping from a temperature of 45 °C to 280 °C, and in both scan and selected ion monitoring (SIM) mode.") in line 230 of the original manuscript version. Therefore, we would prefer not to repeat the explanation of the abbreviation.

*[no change made]*

**ll. 339/340: does that mean, the PTR-MS was calibrated only once, due to the complication of focusing on the oxygenated species? While it makes a lot of sense to calibrate these species individually, I suggest to use gas-standards with e.g. a set of hydrocarbons, ketones, siloxanes (…) to allow for more frequent calibrations to monitor variations in its transmission curve.**

For toluene, the PTR-ToF-MS was calibrated using the mean signal of the start concentration across all experiments. We also have a gas-standard for PTR-MS calibrations, but we preferred to use the chamber calibration as it better resembles the experimental conditions. We did not observe significant variations in the response over the experimental time frame. We tried to formulate this procedure more explicitly and hope that this change will enhance clarity.

Old version (3.2.2, third paragraph):
*For toluene, we evaluated the measured signals in the filled chamber after equilibration and prior to the ignition of the solar simulator across all experiments (Sect. 3.3). These experiments were carried out over a period of 6 weeks, during which time the sensitivity of the instrument is not susceptible to substantial drift. We calibrated toluene using the mean signal for its calculated start concentration of c = 3.79×10$^{12}$ molecules cm$^{-3}$ and the blank value of the cleaned chamber. The overall quantification error was 10 %, calculated as propagation of the instrumental error (0.30 %) and the experimental error (10 %, variability of monitored start concentrations).*

New version (3.2.2, third paragraph):
*For the toluene calibration, we evaluated the measured signal in the filled chamber after equilibration and prior to the ignition of the solar simulator. As we always used the same start concentration of toluene, we averaged the measured response across all experiments (Sect. 3.3). These experiments were carried out over a period of 6 weeks. During this time, we did not observe a substantial drift in the sensitivity of the instrument. We performed a two-point calibration of toluene using the mean signal for its calculated start concentration of c = 3.79×10$^{12}$ molecules cm$^{-3}$ and the blank value of the cleaned chamber. The overall quantification error was 10 %, calculated as propagation of the instrumental error (0.30 %) and the experimental error (10 %, variability of monitored start concentrations).*

**l. 370: many teflon chambers are operated continuously under pressure to avoid contamination from outside. Did you determine the effectiveness of cleaning your chamber both with just flushing under pressure vs with evacuation? (same: L 400 → emptying it to 30% of its original volume). I wonder, what the impact of small leaks would be here. Did you perform leak tests, e.g. by spraying acetone outside the teflon chamber and monitoring it inside?**

We would like to highlight the difference between the cleaning procedure and the experimental procedure. For cleaning (l. 370, original manuscript), we evacuated the chamber completely, thereby also actively sucking residual compounds from the walls. We then filled it with zero air and evacuated it again several times. As the blank values in the chamber were always very low (see Fig. 6), we are confident that our cleaning procedure was effective. The actual experiments were performed in batch mode. The statement about not emptying the chamber to less than 30% of its original volume (l. 400, original manuscript) refers to the experimental procedure. Not emptying the chamber completely had the purpose of (i) minimizing wall effects which become more complex when the area-volume-ratio increases, and (ii) avoiding contamination through leaks. We understand that the text was not perfectly clear about this. We therefore improved it by stressing which procedure was applied for cleaning vs for the experiments.

Old version (3.3.1, 1st paragraph):
*Prior to each experiment, we cleaned the chamber and recorded blank values. To dilute and remove remaining impurities, the chamber was filled with zero air and then evacuated three times in total. Zero air was prepared by purifying pressurized synthetic air with a commercial zero air generator (Messer Griesheim GmbH, SL 50). In the first flushing cycle, the solar simulator was ignited to promote the oxidation of potential residuals and their release from the walls. After the cleaning, preliminary chamber blanks were measured by the SPME-GC-MS and the PTR-ToF-MS to confirm the cleanliness of the chamber and to test the instrumental performances.*

New version (3.3.1, 1st paragraph):
*Prior to each experiment, we cleaned the chamber and recorded blank values. To dilute and remove remaining impurities, the chamber was filled with zero air and then fully evacuated three times in total. Zero air was prepared by purifying pressurized synthetic air with a commercial zero air generator (Messer Griesheim GmbH, SL 50). In the first flushing cycle, the solar simulator was ignited to promote the oxidation of potential residuals and their release from the walls. After the cleaning, we filled the chamber with zero air again, and measured preliminary chamber blanks with the SPME-GC-MS and the PTR-ToF-MS to confirm the cleanliness of the chamber and to test the instrumental performances.*

Old version (3.3.1, 4th paragraph):
*The chamber was never depleted to less than 30 % of its initial volume to avoid too high surface-area-to-volume ratios and an increased importance of wall effects. After all the data were collected, the chamber was evacuated and then cleaned three times (zero air filling and subsequent evacuation).*

New version (3.3.1, 4th paragraph):
*During the experiments, the chamber was never emptied to less than 30 % of its initial volume to avoid (i) too high surface-area-to-volume ratios and an increased importance of wall effects as well as (ii) contamination through leaks. After all the data were collected, the chamber was completely evacuated and then cleaned three times (zero air filling and subsequent evacuation).*

**L 379 ff. – not completely clear to me. Is the toluene flushed with the N2 (15mins, 5slpm) into the larger teflon batch chamber or was the N2 flushing used to clean the toluene flask before toluene was injected? In the latter case, how is the toluene entering the batch chamber?**

The first case, toluene was flushed with the $N_2$ into the chamber. We changed the text to make it clearer.

Old version (3.3.1, 2$^{nd}$ paragraph):
*Upon completion of the preparatory work, we added the reagents sequentially. First, we introduced the VOC precursor into the chamber. The pure compound was injected through a septum into a 100 mL round-bottomed flask, which was flushed with $N_2$ as carrier gas (5 SLPM) for 15 minutes.*

New version (3.3.1, 2$^{nd}$ paragraph):
*Upon completion of the preparatory work, we added the reagents sequentially. First, we introduced the VOC precursor into the chamber. The pure compound was injected through a septum into a 100 mL round-bottomed flask, which was then flushed with $N_2$ as pick-up flow (5 SLPM) for 15 minutes.*

**fig. 3 – I suggest to add long names with the short forms in brackets in the figure description**

We followed the suggestion, and added the short forms of the depicted toluene products to the figure caption.

Added text to the caption of Fig. 3:
*[…] Toluene products in this figure include o-cresol (OCR), m-cresol (MCR), p-cresol (PCR), benzyl alcohol (BOH), and benzaldehyde (BAL).*

For consistency, we added analogous statements to the captions of Fig. 10 and Fig. 11 as well.

Added text to the caption of Fig. 10:
*[…] Toluene products in this figure include o-cresol (OCR), m-cresol (MCR), p-cresol (PCR), and benzyl alcohol (BOH).*

Added text to the caption of Fig. 11:
*[…] Toluene products in this figure include o-cresol (OCR), m-cresol (MCR), p-cresol (PCR), benzyl alcohol (BOH), and benzaldehyde (BAL).*

We also added further explanation to the caption of Fig. S10.

Old version Fig. S10:
*Time series of the two ISTDs and PFBHA as the carbonyl reagent are shown.*

New version Fig. S10:
*Time series of the two ISTDs (phenol-$d_6$ = PHE-$d_6$, and acetophenone-$d_8$ = APH-$d_8$) and PFBHA as the carbonyl reagent are shown.*

**l. 480 – You mention that the reactions with NO3 and O3 are negligible due to their small reaction rates. This is slightly confusing, as I thought the experiments were Nox-free and you did not mention that you actively added O3, so are the experiments not O3 and NO3-free?**

We understand the confusion. For the $O_3$- and $NO_3$-related losses, we referred to the experiments in the presence of $NO_x$ that we performed for method development but did not include in this manuscript to evaluate the toluene mechanism (see section 3.3: "In total, we performed 18 experiments at different temperatures and with different initial $NO_x$ mixing ratios for method development purposes. To evaluate the product formation yields and gain mechanistic insight into the toluene chemistry, we focused here exclusively on six $NO_x$-free toluene-OH photooxidation experiments at T = 298±1 K and different degrees of photochemical aging."). We deleted this sentence to avoid confusion and since the experiments with added $NO_x$ are not relevant to the mechanistic evaluation presented in this work.

Deleted text (end of Sect. 4):
*Its rate constants for reactions with $O_3$ and $NO_3$ radicals are about k $<10^{-21}$ molecules$^{-1}$ cm$^3$ s$^{-1}$ and k = 7.8×10$^{-17}$ molecules$^{-1}$ cm$^3$ s$^{-1}$ (IUPAC, 2024), so that these loss processes were irrelevant under all experimental conditions.*

**Fig. 4: the legend could be improved for more clarity. There are e.g. multiple grey lines in different shades of gray, but only 1 line in the legend. Alternatively, decreasing the visibility of the error-minima and maxima traces and just using shading might help**

As suggested, we decreased the visibility of the error minima and maxima and just used shading to improve clarity. We also included the mercury emission lines with a distinct line pattern in the legend.

To improve comprehensibility, we furthermore labelled the panels, and referred to them in the figure caption and the main text accordingly.

*[see Figure 4]*

Old version caption Fig. 4:
*[...] The left figure shows the emission spectrum of the solar simulator (7 HMI lamps with bandpass filter) available from [...]. In the right figure, the solar simulator spectrum is compared against the natural actinic flux as calculated [...]*

New version caption Fig. 4:
*[...] (a) The emission spectrum of the solar simulator (7 HMI lamps with bandpass filter) available from [...]. (b) The solar simulator spectrum compared against the natural actinic flux as calculated [...].*

Old version main text (sect. 4.1.1):
*The measured emission spectrum of the solar simulator is shown in Fig. 4. [...] For comparison, Fig. 4 shows the resemblance between the solar simulator emission spectrum and the natural solar spectrum, [...]*

New version main text (sect. 4.1.1):
*The measured emission spectrum of the solar simulator is shown in Fig. 4a. [...] For comparison, Fig. 4b shows the resemblance between the solar simulator emission spectrum and the natural solar spectrum, [...]*

**l. 523: how did you find the OH concentration? Was it measured via the toluene decay?**

Yes, we calculated the OH radical concentration via the decay of toluene. To make this clearer, we added more explanation to the calculation of the OH exposure in the section about the loss corrections. We also specifically added the calculation of the OH radical concentrations.

Old version (end of sect. 3.3.3):
*To treat the reaction with OH radicals using the same correction formula, we converted the second order reaction of any of the analytes with OH radicals to a pseudo first order approach, as shown in Eq. (3):*

$$\Delta t \times k' = \Delta t \times k \times [OH] = \Delta OHexp \times k \qquad (3)$$

*where $k'$ is the pseudo first order rate constant in $s^{-1}$, $k$ is the second order rate constant in molecules$^{-1}$ cm$^3$ s$^{-1}$, $[OH]$ is the OH radical concentration in molecules cm$^{-3}$, and $\Delta OHexp$ is the difference of the OH exposure between the two data points which is given in the units of molecules cm$^{-3}$ s. The second order rate constants for the reactions of the photooxidation products with OH radicals were obtained from the IUPAC recommendations and the scientific literature (Table S7). The calculation of the OH exposure at a given point in time was based on the observed decay of toluene to circumvent the need for direct OH radical measurements, as shown in Eq. (4):*

$$\Delta OHexp = OHexp_t - OHexp_{t-1} \qquad (4a)$$

$$OHexp_t = ln\left(\frac{[toluene]_t}{[toluene]_0}\right)/-k_{toluene+OH} \qquad (4b)$$

*where $OHexp_t$ and $OHexp_{t-1}$ are the OH exposures determined for the time steps associated with the target data point and the previous data point in molecules cm$^{-3}$ s, $[toluene]_t$ and $[toluene]_0$ are the concentrations of toluene at the given point in time and at the beginning of the experiment in molecules cm$^{-3}$, and $k_{toluene+OH}$ = 5.6×10$^{-12}$ molecules$^{-1}$ cm$^3$ s$^{-1}$ is the well-characterized second order rate constant of toluene with OH radicals (IUPAC, 2024).*

New version (end of sect. 3.3.3):
*To treat the reaction with OH radicals using the same correction formula, we converted the second order reaction of any of the analytes with OH radicals to a pseudo first order approach. For this purpose, we referred to the OH exposure as the time-integrated OH radical concentration, as shown in Eq. (4):*

$$OHexp_t = \int_{t=0}^{t=t}[OH]\Delta t \qquad (4a)$$

$$\Delta OHexp = \int_{t=t-1}^{t=t}[OH]\Delta t \qquad (4b)$$

$$\Delta t \times k' = \Delta t \times k \times [OH] = \Delta OHexp \times k \qquad (4c)$$

*where $OHexp_t$ is the OH exposure between time t and the beginning of the experiment in units of molecules cm$^{-3}$ s, $[OH]$ is the OH radical concentration in molecules cm$^{-3}$, $\Delta OHexp$ is the difference of the OH exposure between the two data points in molecules cm$^{-3}$ s, $k'$ is the pseudo first order rate constant in $s^{-1}$, and $k$ is the second order rate constant in molecules$^{-1}$ cm$^3$ s$^{-1}$. The second order rate constants for the reactions of the photooxidation products with OH radicals were obtained from the IUPAC recommendations and the scientific literature (Table S7). To*

*circumvent the need for direct OH radical measurements, the OH exposure at a given point in time was calculated based on the observed decay of toluene, as shown in Eq. (5):*

$$\Delta OHexp = OHexp_t - OHexp_{t-1} \tag{5a}$$

$$OHexp_t = \ln\left(\frac{[toluene]_t}{[toluene]_0}\right)/-k_{toluene+OH} \tag{5b}$$

*where $OHexp_t$ and $OHexp_{t-1}$ are the OH exposures determined for the time steps associated with the target data point and the previous data point in molecules cm$^{-3}$ s, $[toluene]_t$ and $[toluene]_0$ are the concentrations of toluene at the given point in time and at the beginning of the experiment in molecules cm$^{-3}$, and $k_{toluene+OH}$ = 5.6×10$^{-12}$ molecules$^{-1}$ cm$^3$ s$^{-1}$ is the well-characterized second order rate constant of toluene with OH radicals (IUPAC, 2024). Based on the temporally resolved OH exposure (available in minute resolution according to the toluene data set), we used the relationship in Eq. (4b) to calculate OH radical concentrations averaged over the time intervals between t and t-1 according to Eq. (6):*

$$[OH] = \Delta OHexp/\Delta t \tag{6}$$

Since we added equation 6, we needed to change the numbering of the following equations accordingly.

*[see following equations]*

**L. 525: that's good! But I believe, this needs to be checked before future experiments after NOx has been added, again.**

We agree. As reviewer 1 also found the statement concerning HONO too speculative, we deleted this part entirely, so as to avoid false implications.

Old version (Sect. 4.1.1, OH radical paragraph):
*We did not observe any product formation in a test run without $H_2O_2$ addition, confirming $H_2O_2$ photolysis as the main source of OH radicals, rather than for example the release of HONO from chamber walls which would produce both OH radicals and NO.*

New version (Sect. 4.1.1, OH radical paragraph):
*We did not observe any product formation in a test run without $H_2O_2$ addition, confirming $H_2O_2$ photolysis as the main source of OH radicals.*

**Fig. 7: "relative response" is not clear from the figure or figure description (only after reading the text). It is not described, in relation to what the signals are shown. Please add units for more clarity.**

We agree that the notion "relative response" is not clear just from the figure. We added the information "area analyte / area internal standard" to the y axis label. We also added the unit of the slope m ("molecules$^{-1}$ cm$^3$").

*[see Figure 7]*

**Fig. 8: please add the shortforms (e.g. PHE-d6) in the figure caption**

We added the shortforms in the figure caption as suggested.

Original version caption Fig. 8:
*Each individual marker represents one measurement of phenol-$d_6$ or acetophenone-$d_8$, respectively.*

New version caption Fig. 8:
*Each individual marker represents one measurement of phenol-$d_6$ (PHE-$d_6$) or acetophenone-$d_8$ (APH-$d_8$), respectively.*

**Fig. 11: instead of "final concentration", maybe call it total produced concentration or similar? And mark it in the plot as a horizontal bar with uncertainties (cause the loss rates also have uncertainties)**

We agree that the term was imprecise. We replaced "final concentration" with "loss-corrected concentration", as we used this term throughout the paper.

Change in the main text:

Old version (beginning Sect. 4.3.1):
*The contribution of the three characterized loss processes (wall losses, photolysis, secondary OH radical reactions) to the final concentration differed [...]*

New version (beginning Sect. 4.3.1):
*The contribution of the three characterized loss processes (wall losses, photolysis, secondary OH radical reactions) to the loss-corrected concentration differed [...]*

Change in the caption of Fig. 11:

Old version:
*Contributions of the non-corrected data and each of the three included loss corrections (wall losses, photolysis, secondary OH radical reactions) to the final concentration. [...]*

New version:
*Contributions of the non-corrected data and each of the three included loss corrections (wall losses, photolysis, secondary OH radical reactions) to the loss-corrected concentration. [...]*

As suggested, we also specifically plotted the loss-corrected mixing ratios and the associated uncertainties in Figure 11. We furthermore labelled panel a and b to increase comprehensibility and adjusted the caption accordingly.

*[see Figure 11]*

Old version of caption:
*[...] The shares of the raw data and the losses are shown on an absolute scale (mixing ratios, upper graph) and on a relative scale (normalized to the total corrected mixing ratio, lower graph). [...].*

New version of caption:
*[...] The shares of the raw data and the losses are shown on (a) an absolute scale (mixing ratios) and (b) on a relative scale (normalized to the loss-corrected mixing ratio). [...]*

**Fig. 12: please just write mixing ratio instead of the new shortform "MR"**

We followed the suggestion and used the full notation "mixing ratio". For consistency, we made the same adjustment in Fig. 13.

Please note that we also fixed a mistake in Fig. 12. Previously, two lines were shifted to a wrong position during the rendering of the plot.

*[see Figures 12 and 13]*

**fig. S6.4: is m your calibration factor? Y-axis unit missing…**

We added the unit "area" to the y axis label. We also added the unit of the slope m ("area molecules$^{-1}$ cm$^3$").

*[see Figure S11 in Section S6.4]*

Further note

As part of this manuscript, we present a weighted calibration for the mass *m/z* 109.0626 at the PTR-ToF-MS, assigned to the chemical structure $C_7H_8O$. In the original preprint, we considered the three cresol isomers as well as benzyl alcohol. During the review of the preprint, we improved the assignment. As the PTR-ToF-MS does not show any signal for benzyl alcohol at this mass due to fragmentation, we decided that the inclusion of benzyl alcohol within the weighted calibration can be misleading. In particular, the difference between the averaged and weighted calibration factor is disproportionately large when including zero-sensitivity compounds. Therefore, we suggest the following modifications:

- to change the calculation and interpretation of the weighted PTR-ToF-MS calibration to include only the three cresol isomers, using their individual sensitivities and relative abundances (0.74:0.04:0.22 for the *o/m/p* isomer, respectively).
- to compare the sum signal at *m/z* 109.0626 against the sum of the three cresol isomers (no longer including benzyl alcohol) at the SPME-GC-MS accordingly (Fig. 9).
- to not evaluate and show any PTR-ToF-MS time series of benzyl alcohol (Fig. 10), as (i) it is not measurable at *m/z* 109.0626 and would no longer be represented in the weighted calibration, and (ii) we prefer not to analyse the unspecific *m/z 91.0522* fragment.

While the PTR-ToF-MS cresol data is not a major part of the manuscript, it is nevertheless addressed in a few different sections. We suggest to make all necessary changes for consistency, as outlined in the following:

Abstract:

Old version:
*For the cresols and benzyl alcohol, we compiled a weighted calibration factor for the PTR-ToF-MS, taking into account isomer-specific sensitivities as well as the relative distribution as determined by the SPME-GC-MS. The weighted calibration improved the instrumental agreement to 15 %, whereas the PTR-ToF-MS overestimated the sum of the isomers by 25 % compared to the SPME-GC-MS concentrations when using the averaged calibration factor.*

New version:
*For the cresols, we compiled a weighted calibration factor for the PTR-ToF-MS, taking into account isomer-specific sensitivities as well as the relative distribution as determined by the SPME-GC-MS. The weighted calibration improved the instrumental agreement to 14 %, whereas the PTR-ToF-MS overestimated the sum of the isomers by 31 % compared to the SPME-GC-MS concentrations when using the averaged calibration factor.*

Methods (last paragraph 3.2.2):

Old version:
*[…] All compounds were analysed at the protonated mass of m/z 109.0626, yet the instrumental response of each of the isomers was derived individually. For analyzing the sum signal during the photooxidation experiments, we calculated the weighted sensitivity for each of the isomers as the product of its instrumental sensitivity and its relative abundance. The relative abundances*

*were derived as fixed values from the formation yields as determined with the SPME-GC-MS data (Sect. 4.3.2). [...] From the correctly quantified sum signal, we later extracted the experimental concentrations of the individual $C_7H_8O$ isomers using the same fixed relative abundances.*

New version:
*[...] All compounds were analysed at the protonated mass of m/z 109.0626, yet the instrumental response of each of the isomers was derived individually. For analyzing the sum signal during the photooxidation experiments, we calculated the weighted sensitivity for each of the isomers as the product of its instrumental sensitivity and its relative abundance. Since benzyl alcohol did not have any signal at this mass (Table S6), only the cresols isomers were included in the weighted calibration. Their relative abundances were derived as fixed values from the formation yields as determined with the SPME-GC-MS data (Sect. 4.3.2). [...] From the correctly quantified sum signal, we later extracted the experimental concentrations of the individual cresol isomers using the same fixed relative abundances.*

Results (section 4.2.3):

➤ 2nd paragraph:

Old version 2nd paragraph:
*[...] Benzyl alcohol did not show any signal on m/z 109.0626 and thus did not contribute to the sum signal. Instead, it fragmented to m/z 91.0522, which is in accordance with the literature (Yeoman et al., 2021) and probably indicates the formation of the benzyl radical after abstraction of water from the protonated molecule. The isomer-specific calibration was performed with relative abundances of 0.71 for o-cresol, 0.04 for m-cresol, 0.21 for p-cresol, and 0.04 for benzyl alcohol (Sect. 4.3.2). As o-cresol is by far the most abundant isomer, its contribution to the weighted calibration factor on m/z 109.0626 was as high as 87 %. Overall, considering the relative abundances and the specific sensitivities of the individual isomers led to a 1.47-fold increase of the calibration factor compared to the non-weighted calibration (Table S6).*

New version 2nd paragraph:
*[...] Benzyl alcohol did not show any signal on m/z 109.0626 and thus did not contribute to the sum signal and was excluded from the weighted calibration. Instead, it fragmented to m/z 91.0522, which is in accordance with the literature (Yeoman et al., 2021) and probably indicates the formation of the benzyl radical after abstraction of water from the protonated molecule. As this fragment is unspecific, we did not evaluate it further. The isomer-specific calibration was performed with relative abundances of 0.74 for o-cresol, 0.04 for m-cresol, and 0.22 for p-cresol (Sect. 4.3.2). As o-cresol is by far the most abundant isomer, its contribution to the weighted calibration factor on m/z 109.0626 was as high as 87 %. Overall, considering the relative abundances and the specific sensitivities of the individual isomers led to a 1.15-fold increase of the calibration factor compared to the non-weighted calibration (Table S6).*

➤ Figure 9:

In Figure 9, the first panel now shows a comparison of the sum of the three cresol isomers, not of all $C_7H_8O$ isomers, changing both the x and y data. The caption was adapted accordingly.

*[See Fig 9]*

[revised manuscript text omitted]

Supplement:

➢ Table S6

In the table, we differentiated the cresol isomers and benzyl alcohol more explicitly, showing plainly that benzyl alcohol is not incorporated into the weighted calibration. We still list its zero sensitivity at *m/z* 109.0626, as we believe this to be of interest to the community. For the weighted calibration, we updated all the values. Note that also the non-weighted values now appear different, as we performed the isomer-specific calibrations at a slightly later point and scaled them to the previously recorded sum signal, which is now composed of three instead of

four compounds. We also changed the caption of the table, specifying the relative abundances as well as changing the mentions of the $C_7H_8O$ isomers to only the cresol isomers.

*[see Table S6]*

Old version caption:
*[...] For the $C_7H_8O$ isomers, we derived a weighted sensitivity by multiplying the recorded sensitivity with the relative abundance as determined by SPME-GC-MS. The weighted calibration factor for the sum signal was obtained from the sum of these weighted isomer-specific sensitivities. When evaluating individual $C_7H_8O$ isomers, we adopted the error and LOD of the weighted sum of the isomers.*

New version caption:
*[...] For the cresol isomers, we derived a weighted sensitivity by multiplying the recorded sensitivity with the relative abundance as determined by SPME-GC-MS (0.74 for o-cresol, 0.04 for m-cresol, and 0.22 for p-cresol). The weighted calibration factor for the sum signal was obtained from the sum of these weighted isomer-specific sensitivities. When evaluating individual cresol isomers, we adopted the error and LOD of the weighted sum of the isomers.*

➢ Section S7

In Section S7, we changed the wording in the introductory text and the caption of Figure S14, and updated the displayed PTR-MS data in Figure S14.

*[see Fig. S14]*

Old version text:
*[...] for o-cresol as one of the $C_7H_8O$ isomers [...]*

New version text:
*[...] for o-cresol as one of the cresol isomers [...]*

Old version caption:
*[...] The mixing ratio of o-cresol at the PTR-ToF-MS was calculated from the weighted calibration and the fixed relative abundance of the $C_7H_8O$ isomers (0.71 for o-cresol).*

New version caption:
*[...] The mixing ratio of o-cresol at the PTR-ToF-MS was calculated from the weighted calibration and the fixed relative abundance of 0.74 for o-cresol.*

---

## Author Comment (AC2)

We thank the referee for the critical review and suggestions, which are much appreciated. The specific comments are addressed one by one in the following. The corresponding changes in the manuscript are highlighted in a marked-up version accordingly. Here, we structure our responses in the following scheme:

**Referee's comments/questions in bold font.**
Author's response in regular font.
*Proposed changes in italic font.*

**Section 4.1.2. The authors parameterize measured wall losses using simple equations. How does this parameterization compare against theoretical considerations or other efforts to quantify wall losses using chemical information? There are a number of papers that have investigated vapor wall loss and developed a number of parameterizations (e.g., Yeh and Ziemann, Ye et al. 2016, McVay et al. 2014 to name a few). These are estimated using C\*, which in itself is related to molecular weight by the ideal gas law. How do these relate to the functions described here? And why the square root of the vapor pressure? Some context and description of previous parameterizations would be helpful.**
**I can see how this is a valid approach for the toluene system, but is it the intention of the authors to use this more broadly for other systems? If so, I would think that it may be necessary to discuss this in more detail and how this parameterization compares to other forms (as noted in comment above). This parameterization seems very simple compared to previous efforts to characterize wall loss rates, but I may not be aware of all of the various efforts.**
**https://www.tandfonline.com/doi/full/10.1080/02786826.2016.1195905**
**https://www.tandfonline.com/doi/full/10.1080/02786826.2015.1068427**
**https://pubs.acs.org/doi/10.1021/es502170j**

Our intention is not to present a mathematical wall loss model, but rather to parameterize the wall losses under our studied conditions and for the toluene photooxidation system. We selected the parameters to rely on easily accessible input-values for straight-forward usage. Following your suggestions, we added an explicit explanation and justified that the reason of the square root was to improve the fit of the parameterization. Furthermore, we made the link between our parameters and the previous studies more explicit.

Old version (section 4.1.2, 2nd paragraph):
*The dependence of the measured wall loss rates of the studied oxygenated compounds on their molecular properties is shown in Fig. 5. The most influential parameter was the vapor pressure, which has already been shown previously to be inversely related to wall losses in Teflon chambers (Yeh and Ziemann, 2015; Zhang et al., 2015). In addition, we found a dependency on the molecular weight of the compounds, which is similar to other studies where partitioning to walls increased with the carbon number of the molecule (Matsunaga and Ziemann, 2010; Yeh and Ziemann, 2015). Finally, the wall loss was also influenced by the degree of functionalization and oxygenation (Lumiaro et al., 2021; Matsunaga and Ziemann, 2010; Yeh and Ziemann, 2015), considered in this study in terms of the oxygen-to-carbon ratio (O:C ratio). Here, we propose a parameterization for the wall loss rate ($R^2$ = 0.80) as given in Eq. (5):*

New version (section 4.1.2, 2nd paragraph):
*The dependence of wall losses on molecular properties has been investigated and described for other Teflon chambers by sophisticated gas-wall partitioning models, which are for example based on the ideal gas law or on related parameters such as the saturation concentration (e.g. Matsunaga and Ziemann, 2010; Ye et al., 2016; Yeh and Ziemann, 2015; Zhang et al., 2015). These models also reveal fundamental relationships, such as an inverse link between wall loss rates and vapor pressure, and an increase in wall losses with increasing carbon number and molecular size as well as with an increasing degree of functionalization. Building onto these findings, we developed a simple parameterization that is based on readily accessible parameters and expresses the wall losses of the toluene photooxidation products as first order processes with a fixed rate constant. We tested several fundamental molecular properties alone*

*and in combination for their ability to describe the empirical loss rates. The most influential parameter was the vapor pressure. We obtained the best fit ($R^2$ = 0.80) when also including the molecular weight and oxygen-to-carbon ratio (O:C ratio) and reducing the impact of the vapor pressure by taking its square root (Fig. 5). We therefore propose the parameterization for the wall loss rate given in Eq. (7):*

Based on another comment by reviewer 2, we also included a statement describing the limitations of the presented parameterization with regard to ambient conditions (temperature, humidity). There, we also addressed the experimental duration, which is short in our cases and therefore enables us to assume fixed first order loss rates. In the „best practice" section that we added based on a suggestion by reviewer 2, we also stress the need to re-assess wall losses for other VOC precursors, longer experiments, and changing conditions.

Added text (end of section 4.1.2)*:*
*It should be noted that the presented wall loss rates as well as their parameterization are valid only for dry conditions and a temperature of 298 K, but may differ if experimental conditions change (Grosjean, 1985; Zhang et al., 2015). Furthermore, both the time frame over which we determined the wall losses empirically and the conducted photooxidation experiments were limited to three to four hours. On longer time scales, desorption processes, wall saturation, and equilibrium between the gas phase and the chamber walls need to be taken into account more specifically (Yeh and Ziemann, 2015; Zhang et al., 2015).*

Added text (within new section 4.1.4):
*Wall losses need to be re-assessed (i) for experiments operated at different temperatures, (ii) for experiments performed in humid air, (iii) for new VOC photooxidation systems with different compound structures and properties, (iv) for new chambers when the Teflon body was replaced (see below), and (v) for longer periods of time and more complex wall interactions if the experimental duration is increased.*

Minor comments

**Lines 101-112: This is a very nice discussion about the utility of derivatization. Can the authors point to some relevant studies on the aromatic systems that have used derivatization to quantify the yields of aldehydes, acids, etc.? I think this would be helpful to place in context why derivatization is a powerful technique and relevant to the studied toluene system.**

We added a few examples to show more specifically in which kinds of studies these derivatization techniques have been used.

Old version (Introduction, 2nd-to-last paragraph):
*[description of techniques …]. These derivatization techniques can also be used in combination to cover a broad range of compounds with diverse functionalities (Pindado Jiménez et al., 2013; White et al., 2014; Yu et al., 1998).*

New version (Introduction, 2nd-to-last paragraph):
*[description of techniques …]. These derivatization techniques have been used, for example, to analyze gaseous and particulate oxygenated products formed in the photooxidation of aromatic compounds (e.g. Gómez Alvarez et al., 2007; White et al., 2014), monoterpenes (e.g. Jang and Kamens, 1999; Yu et al., 1998), or VOC mixtures (e.g. Pindado Jiménez et al., 2013) in laboratory studies. By combining silylation and oxime formation, a broad range of compounds with diverse functionalities can be covered (Pindado Jiménez et al., 2013; White et al., 2014; Yu et al., 1998).*

For the sake of consistency, we did the same for the PTR-ToF-MS part in the introduction.

Added sentence (Introduction, 4th paragraph):
*PTR-MS is selective to compounds with higher proton affinity than water and commonly applied to quantify airborne VOC and their photooxidation products (e.g. Müller et al., 2012; Zaytsev et al., 2019).*

The following paper was added to the bibliography:

*Müller, M., Graus, M., Wisthaler, A., Hansel, A., Metzger, A., Dommen, J., and Baltensperger, U.: Analysis of high mass resolution PTR-TOF mass spectra from 1,3,5-trimethylbenzene (TMB) environmental chamber experiments, Atmospheric Chem. Phys., 12, 829–843, https://doi.org/10.5194/acpd-11-25871-2011, 2012.*

Concerning the derivatization, please note that we improved Figures S6 and S7 by labelling the panels and adapting the captions accordingly.

Old version of captions (Fig. S6 + S7):
*Upper graph: derivatization of […]. Lower graph: typical […]*

New version of captions (Fig. S6 + S7):
*(a) Derivatization of […]. (b) Typical […]*

**Lines 125-132: I would also point towards papers that have studied the SOA and product yields from the second generation products (e.g., cresol, Schwantes et al. 2017). This would be relevant to the discussion that follows about multi-generational formation of glyoxal.**

We added some example studies in which the secondary chemistry of some channels was evaluated, and also added information to the discussion of multi-generational glyoxal formation.

Added sentence after outline of primary chemistry (Sect. 2, end of 1st paragraph):
*The follow-up chemistry of major toluene products such as the cresols, benzaldehyde, or the different dicarbonyls has been the subject of previous laboratory studies (e.g. Atkinson et al., 1980; Liu et al., 1999; Majer et al., 1969; Olariu et al., 2002; Schwantes et al., 2017).*

Old version glyoxal + methylgyloxal part (2nd parapraph Sect. 2):
*The dicarbonyl products glyoxal and methylglyoxal occur not only in the primary chemistry of toluene but also as secondary products in most channels (Atkinson et al., 1980; Bloss et al., 2005; Wagner et al., 2003). Although their primary production is typically dominant (Gómez Alvarez et al., 2007; Volkamer et al., 2001), a contribution from secondary sources to their observed concentrations cannot be fully dismissed.*

New version glyoxal + methylgyloxal part (2nd parapraph Sect. 2):
*The dicarbonyl products glyoxal and methylglyoxal occur not only in the primary chemistry of toluene but are also formed as secondary products in most channels (Atkinson et al., 1980; Bloss et al., 2005; Wagner et al., 2003), especially in the breakdown of other dicarbonyls (Liu et al., 1999). Although primary production is typically dominant (Gómez Alvarez et al., 2007; Volkamer et al., 2001), secondary sources contribute to a fraction of the observed concentrations of glyoxal and methylglyoxal.*

The following papers were added to the bibliography:

*Liu, X., Jeffries, H. E., and Sexton, K. G.: Atmospheric Photochemical Degradation of 1,4-Unsaturated Dicarbonyls, Environ. Sci. Technol., 33, 4212–4220, https://doi.org/10.1021/es990469y, 1999.*

*Majer, J. R., Naman, S.-A. M. A., and Robb, J. C.: Photolysis of aromatic aldehydes, Trans. Faraday Soc., 65, 1846–1853, https://doi.org/10.1039/TF9696501846, 1969.*

**Line 154: This is very impressive temperature range I may have missed this in the text, but what are the dimensions of the containment room? How quickly does it take to reach the desired temperature set point? Is it possible to do dynamic temperature changes during experiments? My take is that this system is about the same size as that described by Osseiran et al. and contained in a similar temperature-controlled vessel. Is that correct? As a note, it might be nice to have a picture of the chamber in a TOC graphic to get a sense of the scale.**
**https://www.sciencedirect.com/science/article/abs/pii/S1001074220301170**

It is true that we did not specify the dimensions of the containment room in the text, and that this would be useful information for the reader. Our setup is quite different than the system described by Osseiran et al., as we do not use a temperature-controlled vessel which is more or less tightly enclosing the chamber. Instead, we placed the chamber into a climatized cabinett which is entirely temperature-controlled and about 3x4x5 m in size. As the room is so much larger than the chamber itself, we feel that this would be difficult to capture in a small picture in a TOC graphic. We added the dimensions of the climatized cabinett to the text.

The temperature set point is reached reasonably fast, depending on the exact set point. During this work, we only ran test experiments between 5 °C and 35 °C, which could all be reached within about half an hour. For -25 °C (the lowest-possible temperature), for sure it would take a longer time. It is not possible to do dynamic temperature changes during experiments, only one set point can be defined.

Old version (beginning Sect. 3):
*The BATCH Teflon chamber was located in a temperature-controlled room (Hans Zettner GmbH), in which the temperature could be adjusted between -25 °C and 35 °C with a stability of ±1 K (DeLonghi, HCS 2550 FTS and Dixell, XR170C).*

New version (beginning Sect. 3):
*The BATCH Teflon chamber was located in a temperature-controlled room (Hans Zettner GmbH, 3×4×5 m), in which the temperature could be adjusted between -25 °C and 35 °C with a stability of ±1 K (DeLonghi, HCS 2550 FTS and Dixell, XR170C).*

We also added the dimensions of the containment room and the chamber itself to the overview graphic of the laboratory setup.

*[see Figure 2]*

**Line 168: Is this the footprint of the Teflon chamber? Can the third dimension be provided to get a sense of the full size at max volume (300 L)?**

Yes, we referred to the footprint of the chamber. We adapted the formulation and changed the structure of the sentence to make this clearer. We also added information about the third dimension as suggested.

Old version (Sect. 3.1):
*The BATCH Teflon chamber was made from UV-transparent fluorinated ethylene propylene (FEP). We constructed the pillow-shaped bag by folding a single sheet of a 50 µm thick FEP film (Chemours, 200A FEP100) and heat-sealing (Dieck, UM 802) the three open sides. Excluding the heat-sealed areas, the empty chamber was 140×110 cm in size. It could be filled to approximately 300 L, in which case the surface-area-to-volume ratio was about SA:V = 0.1 cm$^{-1}$. To reduce the risk of leaks, only one connector was installed at the front side of the chamber and used both as inlet (during filling) and outlet (during ongoing experiments). The chamber was suspended in a metal scaffold, with the two short sides being used as top and bottom ends. It was not further stabilized into a specific shape, so that it remained flexible and fully collapsible, and could be operated in batch mode.*

New version (Sect. 3.1):
*The BATCH Teflon chamber was made from UV-transparent fluorinated ethylene propylene (FEP). We constructed the pillow-shaped bag by folding a single sheet of a 50 µm thick FEP film (Chemours, 200A FEP100) and heat-sealing (Dieck, UM 802) the three open sides. Excluding the heat-sealed areas, the empty and flattened chamber was 140×110 cm in size. It was suspended in a metal scaffold, with the two short sides being used as top and bottom ends. It was not further stabilized into a specific shape, so that it remained flexible and fully collapsible, and could be operated in batch mode. The chamber could be filled to approximately 300 L, in which case the surface-area-to-volume ratio was about SA:V = 0.1 cm$^{-1}$. When it was inflated, it expanded to a maximum of about 100 cm in the third dimension. To reduce the risk of leaks, only one connector was installed at the front side of the chamber and used both as inlet (during filling) and outlet (during ongoing experiments).*

We furthermore added the dimensions of the chamber and the information that it's inflatable to the overview graphic of the laboratory setup, hoping that this improves clarity.

*[see Figure 2]*

**Line 250: Perhaps useful to note the estimated mass that could be collected on the SPME fibers for a typical experiment (e.g., the toluene system)?**

During the calibration, we injected 17 compounds simultaneously into the chamber. For the highest calibration level (5 ppbv), this sums up to a total mixing ratio of 85 ppbv of oxygenated products. But even under these circumstances, we did not see any saturation effect. As the loss of toluene in the experiments is much lower, we expect that the fibre is well capable of collecting the entire product mass formed in the experiments.
We feel that line 250 is too early to address this (e.g. since the calibration has not been described at this point), but we instead elaborated this point a bit more in the results/discussion part (section 4.2.1), where we so far only mentioned the linearity of the calibration curves. We now included the implication for the experiments.

Old version (1st paragraph, Sect. 4.2.1):
*The calibration curves are shown exemplary for the ring-retaining first generation products in Fig. 7. No saturation effect is apparent in these curves, indicating that the calibration interval between c = 0 − 12.3×10$^{10}$ molecules cm$^{-3}$ is still within the linear range of the SPME fibre. [...]*

New version (1st paragraph, Sect. 4.2.1):
*The calibration curves are shown exemplary for the ring-retaining first generation products in Fig. 7. No saturation effect is apparent in these curves, indicating that the calibration interval between c = 0 − 12.3×10$^{10}$ molecules cm$^{-3}$ is still within the linear range of the SPME fibre. As all tested toluene products were injected jointly during the calibration, the total mixing ratio of the products reached about 85 ppbv for the highest calibration level. During the NO$_x$-free photooxidation experiments, the loss of toluene and hence the maximum production of oxygenated compounds (as C$_7$ equivalents) was by about a factor of 4 or more lower than that, so that the sampling capacity of the fibre was likely not exceeded. [...]*

**Line 256-262: Later in the text, the authors note that the stability of ISTDs were tracked for a week using PTR-ToF-MS. I would include that discussion here.**

We moved the sentence "To validate the temporal stability of the permeation source, the enriched outflow was monitored over a period of one week by PTR-ToF-MS." from the Results/Discussion section (4.2.2) to the Method section (3.2.1). With the previous sentence missing, we added some context to the following sentence in the Results/Discussion section.

Old version methods (3.2.1, ISTD parapraph):
*These ISTDs were added into the transfer line between the chamber and the SPME-GC-MS sampling cell by means of a customized permeation source (for details, see Supplement S3.3 and Fig. S5). Phenol-$d_6$ was used to correct [...]*

New version methods (3.2.1, ISTD parapraph):
*These ISTDs were added into the transfer line between the chamber and the SPME-GC-MS sampling cell by means of a customized permeation source (for details, see Supplement S3.3 and Fig. S5). To validate the temporal stability of the permeation source, the enriched outflow was monitored over a period of one week by PTR-ToF-MS. Phenol-$d_6$ was used to correct [...]*

Old version results/discussion (4.2.2):
*In the present setup, the comparability of different samples, of different experiments, and of the experimental period and the on-line calibration was maintained by the added ISTDs. To validate the temporal stability of the permeation source, the enriched outflow was monitored over a period of one week by PTR-ToF-MS. The RSDs of the PI-normalized responses were as low as 5 % for phenol-$d_6$ and 6 % for acetophenone-$d_8$, confirming the suitability of the setup for on-line ISTD addition. Hence, the variability of the ISTD responses measured by SPME-GC-MS (Fig. 8) is indicative for changes in the preparation of the derivatization efficiency, the performance of the GC-MS, and most importantly fibre effects.*

New version results/discussion (4.2.2):
*In the present setup, the comparability of different samples, of different experiments, and of the experimental period and the on-line calibration was maintained by the added ISTDs. The PTR-ToF-MS data showed that the ISTDs in the permeation source outflow were stable over the course of one week, with RSDs of 5 % and 6 % for the PI-normalized responses of phenol-$d_6$ and 6 % for acetophenone-$d_8$ respectively. Hence, the variability of the ISTD responses measured by SPME-GC-MS (Fig. 8) is indicative for changes in the preparation of the derivatization efficiency, the performance of the GC-MS, and most importantly fibre effects.*

**Lines 382-385: Was there an order that experiments were conducted? I.e., were NOx-free experiments conducted prior to those where NO was injected? If not, is there uncertainty associated with possible NOx sources from the walls? Previous studies have shown that walls act as a NOx source once NO or NO2 are introduced as reactants (e.g., Rohrer et al.), so I feel that this should be referenced here. https://acp.copernicus.org/articles/5/2189/2005/acp-5-2189-2005.html**

Yes, the $NO_x$-free experiments were conducted first (see e.g. Figure 8). We agree that this should be mentioned specifically. We therefore added a statement and also included the proposed citation.

Added text (Sect. 3.3):
*[description of evaluated $NO_x$-free experiments …]. These experiments were performed first to avoid the risk for contamination and wall sources of $NO_x$ (e.g. Rohrer et al., 2005).*

Accordingly, the paper was added to the bibliography.

*Rohrer, F., Bohn, B., Brauers, T., Brüning, D., Johnen, F.-J., Wahner, A., and Kleffmann, J.: Characterisation of the photolytic HONO-source in the atmosphere simulation chamber SAPHIR, Atmospheric Chem. Phys., 5, 2189–2201, https://doi.org/10.5194/acp-5-2189-2005, 2005.*

**Line 419: Do wall losses include possible line losses?**

Yes. In the manuscript, we describe how we minimized these losses by heating the transfer line („To maintain comparable conditions between the experiments and to reduce losses, the transfer line from the chamber to the sampling cell (~ 6 m length, 6 mm inner diameter) was temperature-controlled to 50 °C", l. 254-255 in original version). We furthermore expect near-equilibrium between the sample flow and the line walls, as it is not a static system like the chamber interior. We considered such "steady-state line losses" in the calibration, which we performed from the chamber ("To account for losses in the transfer line, the extraction from the sampling cell, and the on-line ISTD addition, we calibrated the analytes determined by SPME-GC-MS directly from within the chamber.", l. 287-288 in original version).

*[no change made]*

**Line 502: For consistency with the previous comparison, I would recommend phrasing that the experimental photolysis rate is "21% higher" than the calculated value.**

We agree that the rewording serves consistency, and therefore implemented the change as suggested.

Original version (Sect. 4.1.1, 1st paragraph):
*[…] is 1.21-fold higher […]*

New version (Sect. 4.1.1, 1st paragraph):
*[…] is 21 % higher […]*

**Line 525-527: This sentence tends to suggest that HONO is not contributing, but this is a little speculative without HONO or measurements of NO / NO2 at low detection limits. I would suggest leaning on the conclusion that H2O2 is the main source of OH radicals and leave out the mention of HONO.**

As suggested we deleted the statement concerning HONO.

Old version (4.1.1, OH radical paragraph):
*We did not observe any product formation in a test run without $H_2O_2$ addition, confirming $H_2O_2$ photolysis as the main source of OH radicals, rather than for example the release of HONO from chamber walls which would produce both OH radicals and NO.*

New version (4.1.1, OH radical paragraph):
*We did not observe any product formation in a test run without $H_2O_2$ addition, confirming $H_2O_2$ photolysis as the main source of OH radicals.*

**Lines 529-531: Here, and elsewhere, it would also be helpful to see the photolysis and OH lifetimes for each species. This would help the reader to gauge how much loss of these species might be expected after they are formed relative to OH or other processes and complement the loss distributions shown in Fig. 11.**

We would like to point out that the rate constants for the wall losses, photolysis rates, and OH radical reactions are listed for each species in the SI (Tables S1, S7, S8) and referenced in the main text accordingly.

However, we understand that a direct comparison and overview would be helpful for the reader. We therefore added a figure showing the absolute and normalized first order rate constants for all three loss processes for each of the compounds in the Supplement (new Figure S12). We prefer showing the loss rates rather than the lifetimes, because for some compounds, absorption cross sections are not available from the literature so that we could not compile photolysis rates (see Table S1). Since in these cases, the loss rate is likely small (meaning that the lifetime is long), it seems to us more intuitive and realistic to exclude them from loss rate plots than from lifetime plots. Also, as conditions change during the experiments, especially with regard to irradiation and OH radical production, we feel that lifetimes could perhaps be misleading.

As additional information to show the extent of the losses (at least for two exemplary species), we included the relative change from the measured to the fully loss-corrected mixing ratios in the other two figures in the Supplement S7 (previously S12 + S13, now S13 + S14).

Added figure in Supplement:

*[see Figure S12]*

Introductory text (in SI, S7):
*The absolute and relative importance of wall losses, photolysis, and OH radical reactions is shown for each of the analysed compounds in Fig. S12.*

Caption Fig. S12:
*Figure S12: Loss processes for each of the analysed toluene photooxidation products. (a) First order loss rates. For the OH radical reactions, we used the mean OH radical concentration of [OH] = 4.89 $\times 10^6$ molecules cm$^{-3}$ for obtaining pseudo first order rates. (b) Normalization of the first order loss rates for each individual compound. Note that we could not compile any photolysis rates for (nitromethyl)benzene (NMB), p-methylcatechol (PMC), and p-hydroxybenzaldehyde (PHB), and assumed zero photolytic loss in these cases (marked with stars). Photolysis is only relevant during chamber irradiation, and the assumed OH radical concentration likewise only applies to the irradiated time period.*

We added the following sentence to the discussion of the losses.

Added sentence (Sect. 4.3.1, 1st paragraph):
*An overview of the first order rate constants of all three loss processes for each of the analysed compounds is provided in Fig. S12.*

Due to the addition of Figure S12, we changed the previous Figures S12 and S13 to Figures S13 and S14, and changed the references in the main text and the Supplement S7 accordingly.

In these figures, we included an additional panel, showing a comparison of the relative change from the measured to the fully loss-corrected mixing ratios for both instruments. We updated the caption, and described the calculation of this relative change in the text above.

Note that we also adapted the colour scheme of Figures S13 and S14 by swapping the purple and black colour to better match the designated colours for the losses in the new Figure S12.

*[See Fig. S13 + S14]*

Main text (4.3.1, last paragraph) old version:
*[…] for different compounds and experiments (see Supplement S7 with Fig. S12 and S13). […]*

Main text (4.3.1, last paragraph) new version:
*[…] for different compounds and experiments (see Supplement S7 with Fig. S13 and S14). […]*

Old version (SI S7):
*In order to evaluate and validate the loss correction procedure at the two instruments, we compared the measured and the loss-corrected data sets obtained by SPME-GC-MS and PTR-ToF-MS. Since the extent of the loss corrections depends on the non-corrected concentrations, we selected experiments in which the measured data of the two instruments agreed particularly well. We show this comparison for benzaldehyde as a compound with no known spectral interference and high photolytic losses (Fig. S12), and for o-cresol as one of the $C_7H_8O$ isomers and a compound with a high reactivity towards OH radicals and a relatively high wall loss rate (Fig. S13). The loss-corrected data are in good agreement.*

New version (SI S7):
*In order to evaluate and validate the loss correction procedure at the two instruments, we compared the measured and the loss-corrected data sets obtained by SPME-GC-MS and PTR-ToF-MS. We show this comparison for benzaldehyde as a compound with no known spectral interference and high photolytic losses (Fig. S13), and for o-cresol as one of the $C_7H_8O$ isomers and a compound with a high reactivity towards OH radicals and a relatively high wall loss rate (Fig. S14). Figures S13 and S14 include not only the time series of the measured data and the individual losses for both instruments (Fig. S13b and S14b), but also the direct comparison of the relative change associated with the loss corrections (Fig. S13a and S14a). This relative change was calculated as the difference between the loss-corrected data and the measured data divided by the measured data, and makes it possible to compare the extent of the loss corrections even in view of slightly different measured initial mixing ratios. The relative changes obtained by the two instruments are in good agreement for both compounds.*

Addition to caption for both Fig. S13 and S14:
*[…] (a) The relative change between the measured and fully loss-corrected mixing ratio for each sample. (b) Absolute raw and corrected mixing ratios. Error bars in both panels are the instrumental quantification error.*

**Figure 8: It is somewhat difficult to see the dark grey bars relative to the lighter grey bars. Perhaps using a different color would help? Or, dotted vertical lines could indicate new experiments?**

We agree that the dark grey and light grey bars were difficult to distinguish. As suggested, we now use dotted vertical lines for new experiments, and additional bars for those experiments where a new fibre was inserted. We adapted the caption accordingly.

*[see Figure 8]*

Old version of caption:
*[…] The grey vertical bars indicate the start of a new experiment. The insertion of a new SPME fibre is illustrated in dark grey. […]*

New version of caption:
*[…] The dotted vertical lines indicate the start of a new experiment. The insertion of a new SPME fibre is illustrated with additional dark grey bars. […]*

**Lines 752-753: This may be the case for chamber studies, but previous studies investigating field observations of isomer distributions (e.g., C8-aromatics, propanal / acetone) have applied isomer-specific sensitivities to interpret PTR signals in very complex mixtures (e.g., biomass burning, Koss et al.). This is also a technique currently used in other mass spectrometers with significant variability in isomer sensitivities (e.g., I-CIMS, Xiong et al. ) https://acp.copernicus.org/articles/15/11257/2015/ https://acp.copernicus.org/articles/15/11257/2015/18/3299/2018/**

The reviewer is correct, and we apologize for this oversight. We deleted the sentence.

Old version (Sect. 4.2.3 last paragraph):
*This is particularly valuable for modelling and for gaining insights into chemical mechanisms. To the best of our knowledge, this study marks the first demonstration of a correction of the sum signal based on both the abundance and the sensitivity of the individual isomers. Rather than performing a separate weighted calibration for each time segment assigned to a specific SPME-GC-MS sample, we used a fixed relative abundance of the $C_7H_8O$ isomers for the weighted calibration and the experimental data.*

New version (Sect. 4.2.3 last paragraph):
*This is particularly valuable for modelling and for gaining insights into chemical mechanisms. Rather than performing a separate weighted calibration for each time segment assigned to a specific SPME-GC-MS sample, we used a fixed relative abundance of the $C_7H_8O$ isomers for both the weighted calibration and the experimental data.*

In the method part, we instead cited the Koss study, but included an equation for our calculation as it was slightly different.

Old version (Sect. 3.2.2 last paragraph):
*To highlight the benefit of the combined analytical instrumentation, we aimed to resolve the $C_7H_8O$ isomers also in the PTR-ToF-MS signal. Therefore, we performed separate calibrations for each of the three cresol isomers and benzyl alcohol. These calibrations were conducted on-line from the BATCH Teflon chamber analogously to the joint PTR-ToF-MS calibration. All compounds were analysed at the protonated mass of m/z 109.0626, yet the instrumental response of each of the isomers was derived individually. For analyzing the sum signal during the photooxidation experiments, we calculated the weighted sensitivity for each of the isomers as the product of its instrumental sensitivity and its relative abundance. The relative abundances were derived as fixed values from the formation yields as determined with the SPME-GC-MS data (Sect. 4.3.2). The sum of these weighted sensitivities was used as the weighted calibration factor for the m/z 109.0626 signal (Table S6). From the correctly quantified sum signal, we extracted the concentrations of the individual $C_7H_8O$ isomers using the same fixed relative abundances.*

New version (Sect. 3.2.2 last paragraph):
*To highlight the benefit of the combined analytical instrumentation, we aimed to resolve the $C_7H_8O$ isomers also in the PTR-ToF-MS signal. In a first step, we compiled a weighted calibration similar to e.g. the work by Koss et al. (2018). Therefore, we performed separate calibrations for each of the three cresol isomers and benzyl alcohol. These calibrations were conducted on-line from the BATCH Teflon chamber analogously to the joint PTR-ToF-MS calibration. All compounds were analysed at the protonated mass of m/z 109.0626, yet the instrumental response of each of the isomers was derived individually. For analyzing the sum signal during the photooxidation experiments, we calculated the weighted sensitivity for each of the isomers as the product of its*

*instrumental sensitivity and its relative abundance. The relative abundances were derived as fixed values from the formation yields as determined with the SPME-GC-MS data (Sect. 4.3.2). According to Eq. (2), the sum of these weighted sensitivities was used as the weighted calibration factor for the m/z 109.0626 signal (Table S6). From the correctly quantified sum signal, we later extracted the experimental concentrations of the individual $C_7H_8O$ isomers using the same fixed relative abundances.*

$$weighted\ calibration\ factor = \sum_i weighted\ sensitivity_i = \sum_i sensitivity_i \times relative\ abundance_i \tag{2}$$

We added the Koss paper to the bibliography:

*Koss, A. R., Sekimoto, K., Gilman, J. B., Selimovic, V., Coggon, M. M., Zarzana, K. J., Yuan, B., Lerner, B. M., Brown, S. S., Jimenez, J. L., Krechmer, J., Roberts, J. M., Warneke, C., Yokelson, R. J., and de Gouw, J.: Non-methane organic gas emissions from biomass burning: identification, quantification, and emission factors from PTR-ToF during the FIREX 2016 laboratory experiment, Atmospheric Chem. Phys., 18, 3299–3319, https://doi.org/10.5194/acp-18-3299-2018, 2018.*

As we added the equation in the method part (Sect. 3.2.2, Eq. 2), we also changed the numbering of the following equations accordingly.

*[see following equations]*

Further note

As part of this manuscript, we present a weighted calibration for the mass *m/z* 109.0626 at the PTR-ToF-MS, assigned to the chemical structure $C_7H_8O$. In the original preprint, we considered the three cresol isomers as well as benzyl alcohol. During the review of the preprint, we improved the assignment. As the PTR-ToF-MS does not show any signal for benzyl alcohol at this mass due to fragmentation, we decided that the inclusion of benzyl alcohol within the weighted calibration can be misleading. In particular, the difference between the averaged and weighted calibration factor is disproportionately large when including zero-sensitivity compounds. Therefore, we suggest the following modifications:

- to change the calculation and interpretation of the weighted PTR-ToF-MS calibration to include only the three cresol isomers, using their individual sensitivities and relative abundances (0.74:0.04:0.22 for the *o/m/p* isomer, respectively).
- to compare the sum signal at *m/z* 109.0626 against the sum of the three cresol isomers (no longer including benzyl alcohol) at the SPME-GC-MS accordingly (Fig. 9).
- to not evaluate and show any PTR-ToF-MS time series of benzyl alcohol (Fig. 10), as (i) it is not measurable at *m/z* 109.0626 and would no longer be represented in the weighted calibration, and (ii) we prefer not to analyse the unspecific *m/z 91.0522* fragment.

While the PTR-ToF-MS cresol data is not a major part of the manuscript, it is nevertheless addressed in a few different sections. We suggest to make all necessary changes for consistency, as outlined in the following:

Abstract:

Old version:
*For the cresols and benzyl alcohol, we compiled a weighted calibration factor for the PTR-ToF-MS, taking into account isomer-specific sensitivities as well as the relative distribution as determined by the SPME-GC-MS. The weighted calibration improved the instrumental agreement to 15 %, whereas the PTR-ToF-MS overestimated the sum of the isomers by 25 % compared to the SPME-GC-MS concentrations when using the averaged calibration factor.*

New version:
*For the cresols, we compiled a weighted calibration factor for the PTR-ToF-MS, taking into account isomer-specific sensitivities as well as the relative distribution as determined by the SPME-GC-MS. The weighted calibration improved the instrumental agreement to 14 %, whereas the PTR-ToF-MS overestimated the sum of the isomers by 31 % compared to the SPME-GC-MS concentrations when using the averaged calibration factor.*

Methods (last paragraph 3.2.2):

Old version:
*[…] All compounds were analysed at the protonated mass of m/z 109.0626, yet the instrumental response of each of the isomers was derived individually. For analyzing the sum signal during the photooxidation experiments, we calculated the weighted sensitivity for each of the isomers as the product of its instrumental sensitivity and its relative abundance. The relative abundances*

*were derived as fixed values from the formation yields as determined with the SPME-GC-MS data (Sect. 4.3.2). [...] From the correctly quantified sum signal, we later extracted the experimental concentrations of the individual $C_7H_8O$ isomers using the same fixed relative abundances.*

New version:
*[...] All compounds were analysed at the protonated mass of m/z 109.0626, yet the instrumental response of each of the isomers was derived individually. For analyzing the sum signal during the photooxidation experiments, we calculated the weighted sensitivity for each of the isomers as the product of its instrumental sensitivity and its relative abundance. Since benzyl alcohol did not have any signal at this mass (Table S6), only the cresols isomers were included in the weighted calibration. Their relative abundances were derived as fixed values from the formation yields as determined with the SPME-GC-MS data (Sect. 4.3.2). [...] From the correctly quantified sum signal, we later extracted the experimental concentrations of the individual cresol isomers using the same fixed relative abundances.*

Results (section 4.2.3):

➢ 2nd paragraph:

Old version 2nd paragraph:
*[...] Benzyl alcohol did not show any signal on m/z 109.0626 and thus did not contribute to the sum signal. Instead, it fragmented to m/z 91.0522, which is in accordance with the literature (Yeoman et al., 2021) and probably indicates the formation of the benzyl radical after abstraction of water from the protonated molecule. The isomer-specific calibration was performed with relative abundances of 0.71 for o-cresol, 0.04 for m-cresol, 0.21 for p-cresol, and 0.04 for benzyl alcohol (Sect. 4.3.2). As o-cresol is by far the most abundant isomer, its contribution to the weighted calibration factor on m/z 109.0626 was as high as 87 %. Overall, considering the relative abundances and the specific sensitivities of the individual isomers led to a 1.47-fold increase of the calibration factor compared to the non-weighted calibration (Table S6).*

New version 2nd paragraph:
*[...] Benzyl alcohol did not show any signal on m/z 109.0626 and thus did not contribute to the sum signal and was excluded from the weighted calibration. Instead, it fragmented to m/z 91.0522, which is in accordance with the literature (Yeoman et al., 2021) and probably indicates the formation of the benzyl radical after abstraction of water from the protonated molecule. As this fragment is unspecific, we did not evaluate it further. The isomer-specific calibration was performed with relative abundances of 0.74 for o-cresol, 0.04 for m-cresol, and 0.22 for p-cresol (Sect. 4.3.2). As o-cresol is by far the most abundant isomer, its contribution to the weighted calibration factor on m/z 109.0626 was as high as 87 %. Overall, considering the relative abundances and the specific sensitivities of the individual isomers led to a 1.15-fold increase of the calibration factor compared to the non-weighted calibration (Table S6).*

➢ Figure 9:

In Figure 9, the first panel now shows a comparison of the sum of the three cresol isomers, not of all $C_7H_8O$ isomers, changing both the x and y data. The caption was adapted accordingly.

*[See Fig 9]*

[revised manuscript text omitted]

Supplement:

➢ Table S6

In the table, we differentiated the cresol isomers and benzyl alcohol more explicitly, showing plainly that benzyl alcohol is not incorporated into the weighted calibration. We still list its zero sensitivity at *m/z* 109.0626, as we believe this to be of interest to the community. For the weighted calibration, we updated all the values. Note that also the non-weighted values now appear different, as we performed the isomer-specific calibrations at a slightly later point and scaled them to the previously recorded sum signal, which is now composed of three instead of

four compounds. We also changed the caption of the table, specifying the relative abundances as well as changing the mentions of the $C_7H_8O$ isomers to only the cresol isomers.

*[see Table S6]*

Old version caption:
*[...] For the $C_7H_8O$ isomers, we derived a weighted sensitivity by multiplying the recorded sensitivity with the relative abundance as determined by SPME-GC-MS. The weighted calibration factor for the sum signal was obtained from the sum of these weighted isomer-specific sensitivities. When evaluating individual $C_7H_8O$ isomers, we adopted the error and LOD of the weighted sum of the isomers.*

New version caption:
*[...] For the cresol isomers, we derived a weighted sensitivity by multiplying the recorded sensitivity with the relative abundance as determined by SPME-GC-MS (0.74 for o-cresol, 0.04 for m-cresol, and 0.22 for p-cresol). The weighted calibration factor for the sum signal was obtained from the sum of these weighted isomer-specific sensitivities. When evaluating individual cresol isomers, we adopted the error and LOD of the weighted sum of the isomers.*

➢ Section S7

In Section S7, we changed the wording in the introductory text and the caption of Figure S14, and updated the displayed PTR-MS data in Figure S14.

*[see Fig. S14]*

Old version text:
*[...] for o-cresol as one of the $C_7H_8O$ isomers [...]*

New version text:
*[...] for o-cresol as one of the cresol isomers [...]*

Old version caption:
*[...] The mixing ratio of o-cresol at the PTR-ToF-MS was calculated from the weighted calibration and the fixed relative abundance of the $C_7H_8O$ isomers (0.71 for o-cresol).*

New version caption:
*[...] The mixing ratio of o-cresol at the PTR-ToF-MS was calculated from the weighted calibration and the fixed relative abundance of 0.74 for o-cresol.*